# Melt Electrospinning of Polymers: Blends, Nanocomposites, Additives and Applications

Anna Bachs-Herrera [1], Omid Yousefzade [1,*] , Luis J. del Valle [1,2] and Jordi Puiggali [1,2,*]

1   Chemical Engineering Department, Escola d'Enginyeria de Barcelona Est-EEBE,
    Universitat Politècnica de Catalunya, c/Eduard Maristany 10-14, E-08019 Barcelona, Spain;
    anna.bh17@gmail.com (A.B.-H.); luis.javier.del.valle@upc.edu (L.J.d.V.)
2   Barcelona Research Center in Multiscale Science and Engineering, Universitat Politècnica de Catalunya,
    Escola d'Enginyeria de Barcelona Est-EEBE, c/Eduard Maristany 10-14, E-08019 Barcelona, Spain
*   Correspondence: omid.yousefzade@upc.edu (O.Y.); jordi.puiggali@upc.edu (J.P.); Tel.: +34-934015649 (J.P.)

**Abstract:** Melt electrospinning has been developed in the last decade as an eco-friendly and solvent-free process to fill the gap between the advantages of solution electrospinning and the need of a cost-effective technique for industrial applications. Although the benefits of using melt electrospinning compared to solution electrospinning are impressive, there are still challenges that should be solved. These mainly concern to the improvement of polymer melt processability with reduction of polymer degradation and enhancement of fiber stability; and the achievement of a good control over the fiber size and especially for the production of large scale ultrafine fibers. This review is focused in the last research works discussing the different melt processing techniques, the most significant melt processing parameters, the incorporation of different additives (e.g., viscosity and conductivity modifiers), the development of polymer blends and nanocomposites, the new potential applications and the use of drug-loaded melt electrospun scaffolds for biomedical applications.

**Keywords:** melt electrospinning; blend and nanocomposite fibers; modifying additives; pharmaceutical applications

## 1. Introduction

Among the new developed techniques for the production of ultrafine 1-dimensional (1D) and 2-dimensional (2D) nanomaterials and non-woven membranes, the best known nanofabrication technology is electrospinning [1–3]. It offers many advantages, such as simplicity and scalability, which are of vital importance in transforming the immense potential of nanoscale materials into useful macroscale assemblies [4,5]. More importantly, compared with analogous nanofiber production methods, such as nanolithography [6], melt fibrillation [7], self-assembly [8], etc., electrospinning offers a unique combination of high production rate, low cost, wide material suitability, and consistent nanofiber quality [9]. Additionally, the technique is versatile and easily adaptable, with variants like needleless and emulsion electrospinning [10], melt electrospinning [11], coaxial electrospinning [12] and co-electrospinning [13] to afford a wide variety of nanoarchitectures like core-shell, tube in tube, porous, hollow, crosslinked, and particle encapsulated structures. Among these techniques, melt electrospinning attracts attention in both academical and industrial environments due to absence of toxic solvent and large scale of fabrication. As schematically shown in Figure 1, the solution electrospinning process is based on mass transfer and evaporation of solvent, resulting in ultrafine fibers, whereas melt electrospinning is performed with heat transfer and quenching of the melted jet. This solvent-free, no-mass-transfer process leads to obtain micrometric fibers, which are wider than those fabricated by solution electrospinning (typically, nanofibers) [14]. However, the high throughput makes it a viable technique for biomedical, energy, and environmental applications. The lack of toxic solvent allows for direct deposition of fiber mats on the chosen substrate and design a

process according to desired final applications [15,16]. Though being a viable process, melt electrospinning presents certain limitations like a large diameter of the electrospun fibers, as mentioned before, as well as a high processing temperature. Therefore, most research efforts with this technique (whilst being less than those involved in solution electrospinning) focuses on the addition of different substances, particles, etc. into the polymeric system to fabricate thinner fibers and provide also specific properties for the different potential applications [11].

Since melt electrospinning is included in the category of polymer processing and is similar to a simple melt spinning, it is necessary to first understand the fundamentals of melt processing of both pure polymers and polymers with additives [17]. Specifically, additives like stabilizers and viscosity modifiers improve melt processing. Moreover, the incorporation of nanocompounds or even the blending with other polymers may drastically enhance physical and mechanical properties of the final products. Nevertheless, cautions must be taken into account to avoid inhomogeneities in the final polymer systems.

The main goal of this review is focused on the recent advances that have been performed on melt electrospinning, considering modifications of the process, selection of materials (e.g., polymers and blends) and discussion of the effects caused by the incorporation of different additives like viscosity modifiers, pharmacological drugs and nanoparticles. Potential applications and future trends of the technique on different areas such as biomedical, filtration and food packaging sectors are discussed/highlighted.

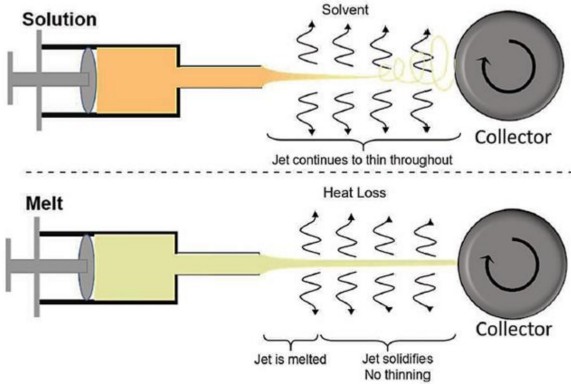

**Figure 1.** Schematic of solution and melt electrospinning processes and how the jets form fibers. Copyright 2018 John Wiley & Sons [14].

## 2. Melt Processing of Polymers with Additives

Pure thermoplastic polymers are seldom processed on their own due to the usual disadvantages that they present including possible degradation, high or low viscosity issues, inability of being processed [18–20] and final applications (e.g., antimicrobial agents can be added to polypropylene to prevent infections [21]). Therefore, pure polymers are compounded with other materials, typically by mechanical blending or melt state mixing, to produce pellets, powders or flakes that are used in subsequent processing operations. The possible additives for thermoplastic polymers include fillers (to reduce cost), reinforcements [22], other polymers [23], stabilizers (to prevent deterioration from light, heat or other environmental factors) [24], various processing aids [25], drugs in biomedical applications [26], etc.

Thermoplastics are usually processed in the molten state, being problems mainly associated to their viscosity and shear thinning behavior. As the shear rate increases, the viscosity decreases, owing to alignments and disentanglements of the long molecular chains. The viscosity also decreases with increasing temperature [27]. In addition to the viscous behavior, molten polymers exhibit elasticity and a number of unusual rheological phenomena. As an example, slow stress relaxation is responsible for frozen-in stresses in the processed products. The normal stress differences are responsible for some flow instabilities during processing and also of the extrudate swell, (i.e., the significant increase

in cross-sectional area when a molten material is extruded out of a die). These properties play an important role in fiber spinning and electrospinning, since elongational forces are applied to fabricate the ultrafine fibers [28,29].

The most important polymer processing operations are extrusion and injection molding, which are material-intensive and labor-intensive processes, respectively. Both these processes involve the following sequence of steps: (a) Heating and melting the polymer, (b) Pumping the polymer to the shaping unit, (c) Forming the melt with the required shape and dimensions, and (d) Cooling and solidification. The suitability of a material for a particular process is usually decided on the basis of the melt flow index (MFI, also called melt flow rate or MFR). This is an inverse measure of viscosity based on a rather crude test involving the extrusion of a polymer through a die of standard dimensions under the action of a prescribed weight. In a simple melt/dry spinning, viscosity or MFI play an important role in both polymer processing ability and final properties of the material [30].

Blends and composites from existing polymers are the preferred form of developing new materials for polymer scientists and industrialists. Lower costs of development, maximum diversification, and the achievement of a broad spectrum of properties are the main reasons. It is also necessary to use the compounding of some virgin plastic with additives to improve the final properties including the reinforcement, electrical insulation characteristics, extension and even polymer processability. The additives used are normally found in powder or in granular form and have different handling and flow characteristics [31]. Nonetheless, some challenges come from the immiscibility and incompatibility between polymers and additives that complicate the processing conditions [31]. Dry blending is useful to have a homogeneous mixture prior to melt processing and compounding [32]. Additives including plasticizers and viscosity modifiers can be mixed with the main polymer in dry state (not molten or solution state) in a high-speed mixer under high shear stress and controlled temperature. Dry blends are generally more economical than molding powders and pellets made by plasticizing and extrusion.

## 3. Conventional Melt Spinning

Melt spinning is, today, one of the most common commercial processes to manufacture synthetic fibers for different applications. The type of the polymer (e.g., amorphous or semi-crystalline) and the processing line, which includes extruder, metering pump and spin pack with spinneret and filter media, are the key factors [32] as shown in Figure 2. The spinneret is a plate with a certain number of holes from which polymeric filaments exit before being cooled down with gas or air. When the polymeric filament leaves the spinneret, it enters a gaseous environment (usually air) at a temperature below the melting point of the polymer. The temperature of the spin line must be below its melting point before reaching any surface to perform elongational flow on the melted polymer [33,34].

By selecting the different materials, their characteristics and also the processing conditions, the final properties of fibers are tunable. The cooling process and drawing ratio lead to control the crystallinity and mechanical properties. If a second stage of further hot-drawing is performed, the chain orientation can be improved, thereby enhancing the properties of the fibers. Besides this hot drawing step, cold drawing is also an useful post process to improve the mechanical properties of thermoplastic fibers since can improve both the orientation and crystallinity of polymers. The effect of fillers and nucleating agents on the final fiber properties has been extensively studied [33,35]. According to the discussion, the electrospinning process from the melt is similar to the simple melt spinning with some differences concerning to the drawing process, cooling and collecting area. Drawing in melt electrospinning is consequence of the application of an electrical force and becomes faster than that obtained by mechanical drawing in a conventional melt spinning. Nevertheless, both of them deal with elongational viscosity, which can be controlled by using additives such as viscosity modifiers. In the next section, different techniques of melt electrospinning are discussed following the concept of melt spinning to manufacture molten jets and collecting systems.

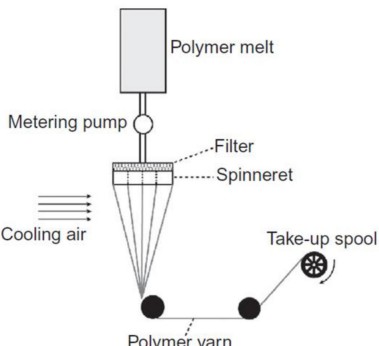

**Figure 2.** Schematic of a typical melt spinning process. Copyright 2015 Elsevier [34].

## 4. Melt Electrospinning

Since melt electrospinning is a solvent-free process, protective equipment that are often used in solution electrospinning can be avoided. Another obvious difference between solution and melt electrospinning devices is the fluid line provided by jets. The plunging system should be in line with the syringe, as fluid lines cool and solidify the polymer melt (see Figure 3), hence the height of the electrospinning device is a significant parameter. For this reason, these vertically configured systems, where the length can be controlled on a typical laboratory bench, may be advantageous.

Syringe pumps are typically used to push the molten polymer. However, alternative configurations have been considered to dispense the polymer melt flow, as shown in Figure 3. The most employed ones are air pressure systems, screw extruders, and mechanical feed. Among the available jet initiation systems in melt electrospinning, needle, needleless and gas-assisted systems are included. Laser can also be used and serves as both the jet initiator and heating source [11]. Shimada et al. [36] developed a system that utilized a line-like laser beam melting device that allowed the generation of multiple Taylor cones on a polymer film.

More detailed information about the technique and process conditions in melt electrospinning is provided by Brown et al. in a review paper [11]. Here we focus on summarizing some new techniques to better understand the performance of materials and additives during the melt electrospinning process, and the influence on the fiber preparation. The different variations focus on the heating systems, the melt of polymers to generate the initial jet and the spin the fiber in continuous way. In addition to this, optimizing the conditions of the process is a key issue to obtain fibers with a desired morphology.

### 4.1. Melt Electrospinning Conditions

Defining the melt electrospinning process conditions mainly depends on the polymer system and the used technique as well as the desired mechanical properties. These properties can undergo a marked improvement compared to typical nonwoven fiber mats in which only the physical entanglement of the fibers contributes significantly to the mechanical properties. The most important parameters in a simple melt electrospinning process are the temperature, tip-to-collector distance, applied voltage, flow rate, and spinneret diameter [16].

Selecting the suitable tip to collector distance relies on the cooling diagram and the ability of stretching the polymer melt. Cooling and consequent solidification of the fibers take place as they stretch, therefore, short tip to collector distances may result in the deposition of slightly molten fibers and in crossover points caused by their fusion [37]. Wang et al. [38] demonstrated that the final fiber diameter not only depends on the distance but also on the collector itself and the temperature. In this study, a KCl/ice water collector (cold collector) proved to produce thinner fibers compared to those obtained with regular tin foils. These thin fibers could have an improved molecular chain orientation and better resistance to the usual shrinkage caused by a reduction of the electric force [37,39,40].

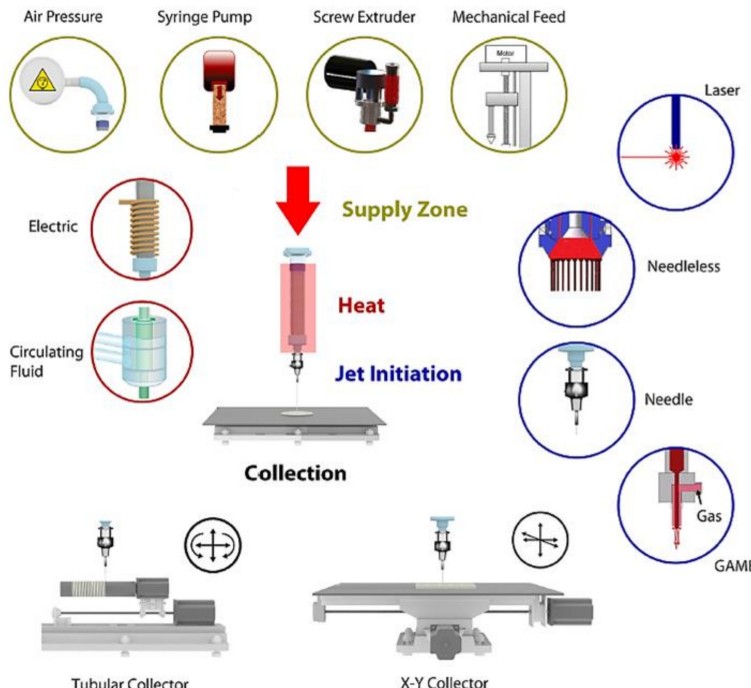

**Figure 3.** Schematics of various heating approaches performed for melt electrospinning. Copyright 2016 Elsevier [11].

Choosing an appropriate processing temperature firstly depends on the melting point of the polymer and then, on obtaining a suitable melt viscosity to stretch the jet, which, as discussed in previous sections, is influenced by temperature. Although it varies among the different techniques, the process temperature is typically chosen between 10 and 30 °C above the polymer melting point. A main drawback of this technique is that melting points are usually so high that polymers may suffer degradation. To avoid this problem, additives can be used so that viscosity is controlled and lower process temperatures can be achieved [41–43].

Influence of different applied voltage in both solution and melt electrospinning follow the same trends reported in a lot of studies [11]. Voltage may range between 7 kV and 50 kV, depending on the technique and material to be electrospun. Polarity (conductivity) and viscosity of the polymer melt are the key properties when it comes to optimize the voltage in order to obtain the desired fiber size and morphology, and get also a stable Taylor cone without sputtering the jets all over the collector and the insulating chamber [17,44].

Another important parameter is the flow rate as it is directly proportional to the fiber diameter (i.e., polymer jets have less time to be stretched at higher flow rates and consequently larger diameters are derived) [45]. The flow rate is remarkably low (between $5 \, \mu L \, h^{-1}$ and $9 \, mL \, h^{-1}$) compared to solution electrospinning due to elimination of the volatile solvent in the initial system. Nonetheless, the production capacity is much more high in the melt electrospinning process as consequence of working with multi jets and non volatile materials [46–48].

The diameter of spinneret holes has logically a great influence on the size of the fiber and may be a limitation to get ultrafine fibers. Since the flow of highly viscous melt polymers through small orifices is challenging and a sufficient flow is necessary to obtain the initial jet, spinneret holes are larger in melt electrospinning than those used in solution electrospinning [37]. Different designs of new spinnerets are now available. For example, non-cylindrical spinnerets could be an option to improve the melt electrospinning technique providing a multi-jet system that could overcome the large-scale barrier [46,47]. On the other hand, in some techniques such as laser or wire melt electrospinning, the spinneret is removed and the initial jets are provided and holed by the solid part of the polymer.

*4.2. New Techniques and Configurations*

Different configurations have been designed and developed to improve the most important steps of the process: melting the material and forming the initial jet. These configurations include the use of an extruder, a ceramic or electrical heater, an oil-circulating heating system, a glass syringe with a heat gun, a gas-assisted system, or a $CO_2$ laser heating system [11]. When choosing one of these, the possible degradation and the control of the viscosity must be considered. In addition, different collector configurations provide different morphologies. As an example, the PCL fiber diameter can be decreased from 2 μm to 270 nm [39] by using the gap method of alignment for collection.

Since the fiber diameter is influenced by the size of initial jet (i.e., the Taylor cone size) and also by the geometrical confinement of the jet, some studies have been focused on finding out their relationship by using simulation and modelling techniques and comparing predictions with experimental results [48,49]. One of the above-mentioned methods, wire melt electrospinning was specially developed for polymers with low melting point like poly (ε-caprolactone) (PCL), a semi-crystalline thermoplastic polyester. In a study carried out by K. Morikawa et al. [14], PCL thin fibers were produced using this method and applying a strong electrostatic field gradient. Figure 4 shows schemes of the most relevant setups— syringe, edge and wire configuration. It is obvious that, among the different methods, the initial jet is much thinner in wire electrospinning, being feasible to get ultrafine fibers (Figure 4c). In this method, a wire is coated with the polymer to generate the thinner initial jet. Taylor cone and size and shape of the initial jet depend on the electrostatic field around the wire tip. The small and narrow region with a size around 200 μm provides high-enough electrostatic forces to overcome the surface tension and lead ultrafine fibers [14]. Recently, a free surface melt electrospinning platform was used. In this case, multiple spontaneous jets were formed from a flat plate (schematically shown in Figure 4b). Solidification point was introduced as a major factor contributing to fiber diameter reduction in an unconfined melt electrospinning. This free surface melt electrospinning technique was found as an effective strategy to control the fiber diameter aiming the solidification point and temperature profile in an electrohydrodynamic jet [50,51].

Among the different configurations and techniques to obtain micro and nanofibers, melt electrowriting (MEW) has been developed to reproduce designed patterns in a similar way to those obtained by 3D printing using computer-aided designs (CAD) [52]. In melt electrowriting, both nozzle and collector are placed in an appropriate position and the extruded filament becomes directly deposited onto the collector following program instructions. The new deposited material is still in the melt state, which makes an efficient crossover with the previous layers before solidification possible. The collector is simultaneously moved in a distance corresponding to the height of one filament, so the layers can be stacked as shown in Figure 5. This layer-by-layer deposition is repeated until the desired 3D object is fabricated [53,54]. In melt electrospinning a separation distance is established between the dispense header and the collector, while in direct electrospinning the extruded polymer is applied directly to the collector surface, or on previous layers of the object under construction. The application of a high voltage allows a greater reduction in the diameter of the filament and therefore the improvement of the resolution of the writing process. Figure 5c,d show representative examples of samples processed by electrowriting.

Li et al. [55] proposed the bubble melt electrospinning technique to prepare polylactide (PLA) and polyurethane (TPU) fibers with characteristics similar to those reported by Liu and He [56] from solution electrospinning. This new needleless technique is based on the formation of bubbles into the polymer melt as consequence of air circulation. These bubbles burst when an electrostatic field is applied between the melt and the collector, causing the generation of polymer jets that deposit onto the collector surface. The number of bubbles and jets that will be produced basically depends on the viscosity of the polymer melt, the gas pressure, the air flow rate, the applied voltage and the distance between the surface where the bubbles are generated and the collector.

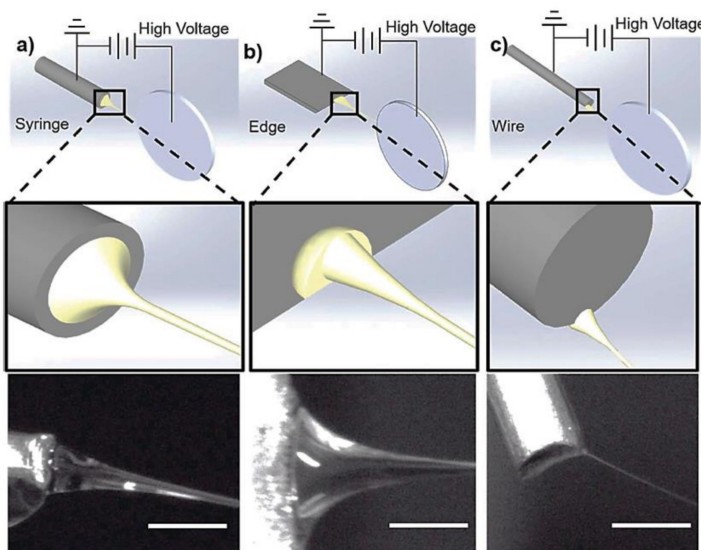

**Figure 4.** Schematic and Taylor cone images for various melt electrospinning methods: (**a**) syringe, (**b**) edge, and (**c**) wire melt electrospinning. In the schematics, grey, yellow and white correspond to the spinneret, the molten polymer and the collector, respectively. All scale bars are 1 mm. Copyright 2018 John Wiley & Sons [14].

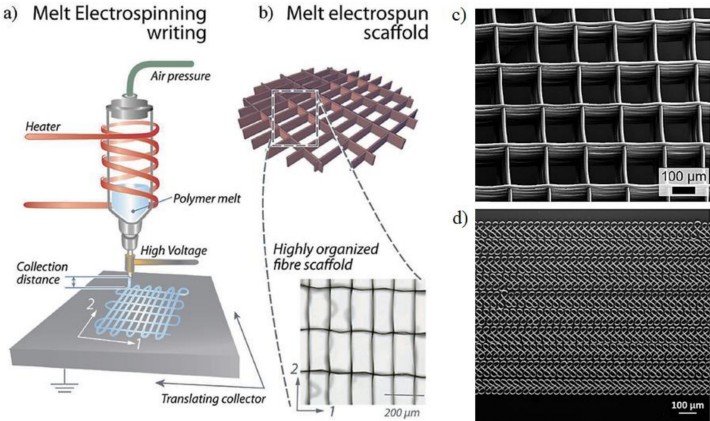

**Figure 5.** (**a**) Schematic of a custom-built MEW device and its principal components: dispensing unit assisted by air-pressure; electrical heating system; high-voltage source electrode; and computer-assisted collector plate. (**b**) Schematic of a melt electrospun fiber scaffold. (**c**,**d**). Micrographs showing different types of melt electrowritten samples. Copyright 2017 John Wiley & Sons [54], Copyright 2017 Elsevier [53].

A gas-assisted melt electrospinning technique has also been developed to prevent polymer degradation and control the nozzle temperature. In this technique, a heated gas vapor is applied to one or more jets. In this way, the solidification time can also be modified, and the drag forces increased due to the high gas flow rate. The method allows achieving a higher production rate and obtaining thinner fibers. Figure 3 shows that the required equipment is simple and consists of the basic melt electrospinning set up equipped with a multi axial gas-jet [57,58].

Among all the techniques and configurations to manufacture thin fibers with a high production rate, there is a novel method for rapid and uniform heating that consists of irradiating the sample with a $CO_2$ laser beam. Many polymers have been successfully electrospun into nano- or microfibers from both rod and sheet preforms as schematically shown in Figure 6 [59,60]. The end of the rod or sheet is melted by the irradiation of laser beams from three directions. The melting process is rapid and consequently can provides a

high production rate and minimize degradation. Parameters such as the feed rate, laser wavelength, diameter of the laser spot and the power of the laser influence the morphology and quality of the obtained fibers [61,62].

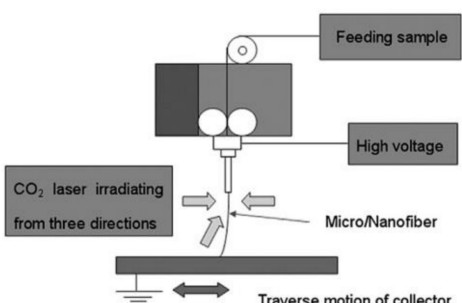

**Figure 6.** Schematic illustration of laser melt electrospinning. Copyright 2012 Elsevier [59].

## 5. Additives in Melt Electrospinning

Despite the above-indicated advantages of melt electrospinning over solution electrospinning [60–62], there are some constraints associated with the process. Some of these challenges are: (a) Large fiber diameter due to the high viscosity of the polymer melts, (b) Complexity of the required equipment, (c) Electric discharge problems associated with the polymer melt, and (d) Intrinsic difficulties caused by the high-temperature setup and the low conductivity of polymer melts [62,63]. To overcome such limiting conditions, different additives can be used depending on the nature of the polymer, the applications, and the developed methods and techniques (see Tables 1–3 as summary).

### 5.1. Salt, Viscosity Modifiers, Stabilizers and Dyes

Isotactic polypropylene (iPP) draws attention both in the fiber industry and research because of its good mechanical properties, excellent chemical resistance, low cost, and polymorphism [64]. Melt electrospinning of iPP was firstly studied by Larrondo in the 1980s [65] and thenceforth, different efforts have been focused on improving the morphology and properties of melt electrospun fibers. Probably, these works are relevant to discuss the effects of rheology modifiers to decrease the viscosity of the medium and facilitate the formation ultrafine and continuous fibers. Basically, iPP cannot provide thin fibers at low voltages, while at high voltages the formation of continuous fibers is hindered due to the strong electrostatic force that causes fibers to burst forth [66].

Ionic salts in both solution and melt electrospinning have been proven to improve the electrical conductivity. The effect of additives (e.g., sodium oleate (SO), poly (ethylene glycol) (PEG), and poly (dimethyl siloxane) (PDMS) at different weight percentages) on the fiber diameter of melt-electrospun PP was investigated by Nayak et al. PEG and PDMS are well-known polymers for lowering the melt viscosity, while SO has the potential to improve the electrical conductivity during melt electrospinning [66]. Therefore, SO can change melt properties, but can also provide uniform and thin fibers (i.e., down to the nanometric scale). SO (7 wt.%) and NaCl (5 wt.%) have also been incorporated [43] leading to nanofibers with diameters of 371 and 310 nm, respectively. Above these concentrations. At higher salt concentrations, melt-electrospinning became unstable causing large Taylor cones and a significant increase of the fiber diameter. Results showed that the electrical conductivity was higher with the addition of NaCl due to its smaller ionic and higher charge density. This feature increased the mobility under the influence of the external field, increased also the stretching of the melt jet and lead to thinner fibers. Addition of SO influenced on the melting process that became more complex and also decreased the hydrophobic character of PP fibers.

Other salts such as stearates of mono and divalent metals including sodium, magnesium, calcium, or zinc were also added to PP to obtain fibers with diameters within the 1–5 μm range (i.e., a significant reduction compared to the diameters larger than 20 μm

that were attained with pure PP processed under similar conditions). It was also determined that the addition of 5 wt.% of calcium, zinc, magnesium and sodium stearates decreased the melt viscosity of pure PP from 230 Pa·s to 210 Pa·s, 179 Pa·s, 138 Pa·s and 107 Pa·s, respectively. It was also found that the electrical conductivity of PP (~9 × 10$^{-12}$ Sm/cm) was increased by 4, 5, and 8 times after addition of calcium, zinc, and magnesium stearates and almost by two orders of magnitude when the same amount (i.e., 5 wt.%) of sodium stearate was incorporated. In this case, the thinnest fibers with diameters around 1.3 μm were obtained [67].

Stearic acid and sodium stearate (SS) have also been added to PP to get melt electrospun matrices for filtration and clothing applications. Both small polar compounds had a plasticizing effect and were efficient to reduce the fiber diameter, provide a uniform size distribution and even modified the thermal properties [67]. Polarization of the carboxyl group of the stearic acid molecule under the electric field lead to a high elongation force and thin microfiber. Smaller diameters were attained with SS since in this case positive and negative ions were able to migrate to opposite ends, giving rise to a strong electric force along the direction of the applied field [67].

Irgastat® P is a usual antistatic additive of PP that has also a positive effect on the electrical conductivity of the polymer melt. Irgastat® P is widely employed in electronics, packaging industry, transportation, in processes such as extrusion and fiber spinning, and logically to avoid dust accumulation [68]. Irgastat® P allows the deposition of a conductive layer with charges homogeneously distributed over the PP matrix with an insulator character. Daenicke et al. [40] added Irgastat® P16 (4 wt.%) and SS (2 wt.%) to obtain an antistatic effect and to increase the electrical conductivity, respectively. Fibers with a 316 nm diameter were produced, a value that could be decreased (i.e., 210 nm) when a low molecular weight PP was added and the amounts of SS and Irgastat® P16 were changed (i.e., 6 wt.% and 2 wt.%, respectively). Experiments were carried out with a processing temperature of 210 °C, a climate chamber temperature of 100 °C, a collector voltage of 45.0 kV and a nozzle-collector distance of 10.0 cm [41].

Cao et al. [69] prepared melt electrospun iPP fibers with diameters smaller than 5 μm by incorporating the β-nucleating agent (WBG). This contains lanthanum and calcium ions [69], induces the crystallization of PP in its β form, has a lubricant effect on the polymer melt and improves mechanical properties, such as toughness and tensile strength. The decrease of the shear stress during processing leads to an improvement of the fluidity of the polymer melt. The process temperature (close to 265 °C) caused physical crosslinking of the fibers and so the formation of a 3D network, with high mechanical properties. The repaired fibrous membrane was suitable for different applications such as protective clothing materials, filtration media, reinforcements for composites, and tissue engineering scaffolds [69].

In a similar way to PP, polyethylene (PE) cannot be processed by a common solution electrospinning process. Melt electrospinning could surmount this problem since no solvent is needed, but again processability would be limited by its high melt viscosity and low conductivity. For this reason, low molecular weight wax and salts were added, being an additional advantage the feasibility to get homogeneously dispersed samples [70,71].

Poly (methyl methacrylate) (PMMA) is a thermoplastic and transparent polymer widely used in optical instruments, electrolytes, aerospace, biomedicine, sensors and solar applications [72]. The plasticizer di-2-ethylhexyl terephthalate (DOTP) has been employed to reduce the viscosity of molten PMMA and facilitate the jet formation in melt electrospinning. Different ratios of PMMA/DOTP were investigated, being concluded that a 40 wt.% of DOTP could provide the thinner fibers (diameter close to 19.7 μm that contrasts with the value of 34.0 μm obtained with the pure polymer). Diameter could be decreased up to 4.0 μm after by optimization of processing parameters ((including the use of a KCl/ice-water solution collector). However, diameters were in all cases far larger than those obtained from solution electrospinning [38].

Low molecular additives, SS, SO and sodium myristate (SM), have also been found as potential additives to improve the properties of polyamide 6 (PA6) melt electrospun fibers. PA6, with excellent physicochemical characteristics, such as good mechanical properties, and heat and chemical resistance for a wide range of chemicals, is used to produce fibrous materials for different applications [73–75]. Pure PA6 has a melting point of 214 °C, but even at 345 °C its high viscosity hinders to get thin fibers by melt electrospinning (i.e., diameters 20 μm are usually obtained). By adding 10 wt.% either of the indicated surfactants, viscosity drops from 5.2 Pa·s to 0.5–0.6 Pa·s independently of the selected salt. Electrical conductivity is also modified in the melt due to their ionogenic character. The increase of the processing temperature is also beneficial since decreases the melt viscosity and increases charge mobility. By controlling and optimizing these parameters, thinner fibers can be obtained (i.e., below 1.5 μm). These fibers could be potentially applied in the filtration of suspended particles from gases, specifically, air. As the average diameter of PA6 fibers decreases, the efficiency of filtration is found to increase significantly. The highest filtration coefficient is observed in the case of materials produced in the presence of 10 wt.% of sodium stearate [76].

PCL fibers have shown great advantages, such as biodegradability and non-toxicity, which make them excellent candidates in tissue engineering and wound dressing. However, the superhydrophobicity of PCL makes it insoluble in nontoxic or scarcely toxic solvents, such as water or dimethyl sulfoxide [77]. Therefore, melt electrospinning makes a great solution to manufacture PCL fibers for biomedical applications such as drug delivery, wound healing and tissue engineering [78]. In these applications, the improvement of the electrical conductivity and the decrease of melt viscosity should logically be performed by adding nontoxic additives. Sodium chloride (NaCl) is an excellent option due to lack of toxicity and easy extraction from the obtained fiber [79]. Piyasin et al. [80] observed that whilst continuous and smooth PCL fibers could be melt electrospun without any additive, their diameter was quite large (~23 μm). On the other hand, the addition of NaCl up to 8 wt.% decreased the size of the fibers down to ~3 μm, but led to slightly rough surfaces and the disruption of the fiber continuity. A higher content of NaCl (i.e., 25 wt.%) had the opposite effect and gave continuous and smooth fibers and a diameter increase up to values similar to those of pure PCL (Figure 7). The melting point was also altered, giving the lowest value at a concentration of 8 wt.% (i.e., 55 °C). This change was attributed to the size of crystalline domains. Curiously, some NaCl was found in the glass syringe after the process, thereby indicating a non-complete ejection through the nozzle.

Salts have also been used as additives in the melt electrospinning of PLA fibers, the most commercially available biobased and biodegradable polymer. This semi-crystalline thermoplastic has a vast number of applications because of its high tensile strength and Young's modulus. Brittleness and high hydrophobicity are the main drawbacks of PLA that may limit, for example, the number of biological applications. Blending PLA with softer polymers or forming composites would be good solutions to improve properties [81]. Koenig et al. [82] studied the effect of NaCl, SS, and a polyester (polyethylene succinate, PES) on PLA properties at both laboratory and pilot scales. PES can be employed as a plasticizer to reduce the viscosity of polyesters (e.g., PLA), PP and polyamides. At laboratory scale, a single-fiber melt-electrospinning device was used, and different temperatures were tested depending on the additive. Melt electrospinning of PLA/NaCl and PLA/PES was performed at 250 °C, whereas PLA/SS was melt electrospun at 190 °C to avoid degradation. Fiber diameter were in the order of 16 to 20 μm for additive contents of 2–4 wt.%. SS was the only additive that provided both thinner fibers and a stable melt-electrospinning process. Therefore, addition of SS was tested at pilot scale considering different concentrations, spin pump speeds, and temperatures. Results pointed out that using a 6 wt.% of SS, an spinneret temperature of 195 °C and a spin pump speed of 0.5 rpm allowed the obtainment of the minimum fiber diameter, which was around 3.8 μm.

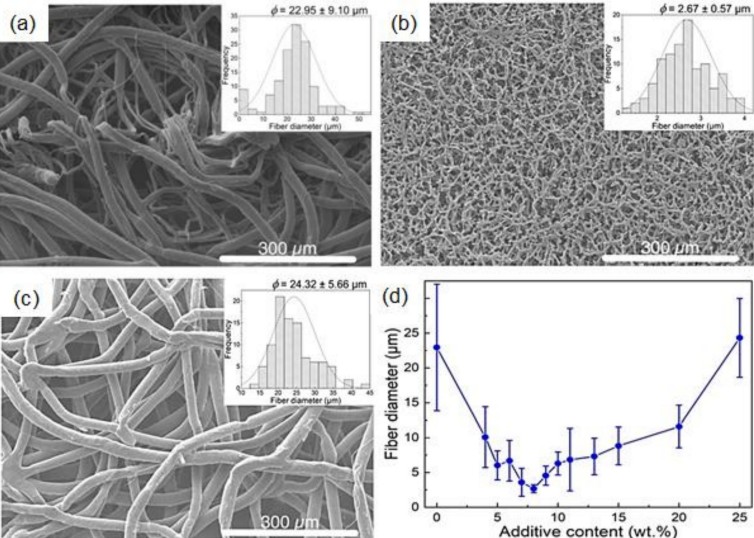

**Figure 7.** Effect of salt additives on the melt-electrospun PCL fibers. The micrographs of the nonwoven materials is: (**a**) without additive, (**b**,**c**) with an NaCl additive of 8 wt.% and 25 wt.%, respectively. (**d**) Average diameter of the PCL fiber as a function of the additive content. Copyright 2019 MDPI [80].

Addition of three biobased dyes (i.e., alizarin, quercetin, and haematoxylin) to increase PLA conductivity has also been evaluated [83]. These environmentally friendly compounds have as main advantage a good solubility/affinity with PLA. Haematoxylin provided the lowest fiber diameters values (i.e., 16.04 µm at 2 wt.%), as consequence of its plasticizing effect and unfortunately for causing a significant polymer degradation. The presence of the indicated additives led to a fivefold decrease of PLA electrical resistance (around 5.0 GΩ), being alizarin the more conductive compound [84].

Qin et al. [85] studied the effect of acetyl tributyl citrate (ATBC) on the diameter of PLA melt electrospun fibers. ATBC is an FDA-approved pharmaceutical plasticizer used in food wrap, vinyl toys, and pharmaceutical excipients [86] that can be employed to decrease the viscosity of the polymer melt. Airflow melt differential electrospinning rendered PLA nanofibers with a diameter as low as 236 nm by incorporating a 6 wt.% of ATBC, and applying an airflow of 25 m/s and a process temperature of 240 °C.

### 5.2. Pharmaceutical Additives

In recent years, there has been an increasing attention for melt electrospinning in the pharmaceutical field as it overcomes certain disadvantages that solution electrospun fibers and other drug delivery systems present, some of which have been previously mentioned in this review. Other hindrances related to solution electrospun fibers are the low loading attained when poor soluble drugs are involved [87]; typical burst release; and low drug in the thin fibers [88,89]. However, it is important to mention that, despite the growing interest in the technique, it is still at the dawning/development step and that most papers being published in this field use solution electrospinning [90]. For this reason, melt electrospinning is not a panacea—or at least, not yet—as there are several aspects that need to be improved.

Currently, studies on melt electrospun fibers able to be used as drug nanocarriers are being developed (see Table 2) [91]. Nevertheless, most biological molecules are thermosensitive, which in some cases make their use in melt electrospinning impossible [92,93]. Logically, the incorporation of plasticizers can be useful in some cases as they decrease the process temperature [94,95] by lowering intermolecular forces among polymer chains [96].

Three Food and Drug Administration (FDA) inactive pharmaceutical ingredients (triacetin, Tween 80, and PEG 1500) have been approved and considered for melt electrospinning. Balogh [95] has successfully melt electrospun Eudragit® E fibers loaded with carvedilol. A good compatibility was observed between Eudragit®, a thermoplastic

amorphous methacrylate terpolymer matrix, and carvedilol [87], a highly lipophilic and thermosensitive beta-blocker drug [97]. Results showed that the application of plasticizers reduced the polymer melting temperature and prevented the degradation of the drug. Furthermore, and the dissolution and release of carvedilol from the fibers was ultrafast, especially in comparison with the pure crystalline drug.

**Table 1.** Salt, viscosity modifiers, stabilizers and dyes as additives in melt electrospinning.

| Polymer | Additive | Effects in the Process * | Fibers Diameter | Ref. |
|---|---|---|---|---|
| PP | Sodium oleate (7 wt.%) | ↑ electrical conductivity, ↓ hydrophobicity | 371 nm | [43] |
| | NaCl (5 wt.%) | ↑ electrical conductivity | 310 nm | [43] |
| | Poly(ethylene glycol) (PEG) | ↓ melt viscosity | | [66] |
| | Poly(dimethyl siloxane) (PDMS) | ↓ melt viscosity | | [66] |
| | Calcium stearate (5 wt.%), Zinc stearate (5 wt.%), Magnesium stearate (5 wt.%) | ↓ melt viscosity, ↑ electrical conductivity | 1–5 μm | [67] |
| | Sodium stearate (5 wt.%) | ↓ melt viscosity | 1.3 μm | [67] |
| | Sodium stearate, Stearic acid | Plasticizer | | [69] |
| | Irgastat® P16 (6 wt.%), Sodium stearate (2 wt.%) | Antistatic (insulator character), ↑ electrical conductivity | 210 nm | [40,68] |
| | β-Nucleating agent ($La^{+3}$ and $Ca^{+2}$) crystallization in β form | ↓ shear stress, ↑ tensile strength, ↑ toughness, ↑ fluidity | <5 μm | [66] |
| PMMA | Di-2-ethylhexyl terephthalate (DOTP) (40 wt.%) | Plasticizer, ↓ melt viscosity (facilitate the jet formation) | 4 μm | [38] |
| PA6 | Sodium stearate, sodium oleate, sodium myristate (10 wt.%) | ↓ melt viscosity, ↑ electrical conductivity | <1.5 μm | [76] |
| PCL | NaCl (8 wt.%) | ↓ melt viscosity, ↑ electrical conductivity | ~3 μm | [80] |
| | NaCl (25 wt.%) | ↑ melt viscosity, ↓ electrical conductivity | ~25 μm | [80] |
| PLA | Sodium stearate (2–4 wt.%) | ↓ melt viscosity | 16–20 μm | [82] |
| | Polyethylene succinate (PES) | Plasticizer, ↓ melt viscosity | | [82] |
| | Haematoxylin (2 wt.%) | Plasticizer, ↑ polymer degradation, ↓ electrical resistance | 16.04 μm | [83] |
| | Acetyl tributyl citrate (ATBC) (6 wt.%) | Plasticizer, ↓ melt viscosity | <236 nm | [85] |

* The arrows ↑ and ↓ indicate increase and decrease, respectively.

Semjonov et al. [98] studied also the effect of water as a plasticizer in the incorporation of indomethacin, another poorly water soluble drug, using the copolymer Soluplus® as drug carrier. Results demonstrated the plasticizing effect of water, the fast drug release and that both polymer and drug remained stable during the spinning process.

The drug dissolution and release from fiber mats do not only depend on the fiber composition [99], but also on the geometry of the final product (thin film, cylindrical sample, etc) [100–102], the fiber diameter and porosity [103,104], the way as the drug was loaded [101,105], and the hydrophilic nature of the matrix [106,107]. In many of the recent studies where drug-loaded polymer fibers were prepared by melt electrospinning, PCL has been selected as the drug carrier due to its excellent properties. Thus, PCL is compatible with other polymers [108], its degradation products can be metabolized by the citric acid cycle [109], it is biocompatible according to both in vitro and in vivo assays [110,111]. Probably, the main limitation of PCL for biological applications concerns

to its superhydrophobicity [112]. Lian and Meng [90] compared the behavior of solution and melt electrospun scaffolds loaded with curcumin, an anticancer agent, which was selected due to its green fluorescence [113] that facilitates observation of its dispersion inside the matrix, and for its high thermal stability [114]. Different concentrations were tested and, whilst it did not affect the morphology of melt electrospun fibers, the drug load was limited in solution electrospinning. Another advantage of the melt electrospun fibers was the lack of a burst release and the low drug release rate. Lian and Meng created a PCL membrane for localized tumour therapy [115], in this case, daunorubicin hydrochloride was selected as the therapeutic agent. Again, different drug concentrations were tested, although some aggregation occurred with increasing drug contents, fiber diameters barely fluctuated and smooth surfaces were always observed. No burst release was detected and the release rate of the drug was again slow. Thus, PCL melt electrospun scaffolds become good candidates for long-term drug delivery systems for tumour therapy.

Among bioactive molecules, antibiotics have also been evaluated in melt electrospun scaffolds. Fabrication of nanocomposite blends based on PCL, with different content of PEG and loaded with ciprofloxacin was performed for wound dressing applications [116]. These fiber mats were successfully fabricated by direct-writing melt electrospinning (Figure 8). PEG properties are excellent for biological applications; hence it is suitable for different functions other than plasticizing. Opposite to PCL, PEG is highly hydrophilic, which enhanced the hydrophilic properties of the fiber mat. It also changed the release mechanism of the drug, making drug diffusion from the mat the main factor in the release. Moreover, release behaviour was observed to change according to the different geometric structures that were tested with the same composition. This means that, as mentioned above, the drug dissolution and release depend on several parameters, including the geometry of the final product and the hydrophilicity, and so this behaviour could be controlled by modifying these two.

Triclosan (TCS) is a synthetic antimicrobial agent that has been largely employed in health and personal care products including deodorants, soaps, toothpaste, mouthwash, and preparations for skin protection [117,118]. TCS has also been loaded in melt electrospun scaffolds of nanocomposite blends composed of PLA, starch, PCL and nanohydroxyapatite (nHAp) [82] taking into consideration previous works concerning the optimization of processing parameters [119–121]. Very abundant in nature, starch is a biodegradable material with high water absorption and weak mechanical properties. In blends, it enhances PLA hydrophilicity [122] and prolongs PCL biodegradation rate [108]. Nanohydroxyapatite (nHAp) is a calcium phosphate similar to natural bone [123] with bioactive properties that is used as bone substitutes and coating for implants because of its osteoconductivity and strong bonding with bone tissue both in vitro [124] and in vivo [125]. Compared to micron-sized ceramics, nHAp shows a greater protein adsorption and osteoblast adhesion [126,127]. nHAp is also a brittle material. This property might limit its applications although its incorporation in polymer matrices could overcome this hindrance. The function of nHAp in this study was to eliminate the adverse effects of triclosan through its encapsulation and increasing the cell attachment. The resulting nanocomposite blend shows improved properties (hydrolytic degradation, hydrophilicity, antibacterial activity, and drug release) with the addition of 3% nHAp with respect to their previous works. Therefore, this could have potential applications in the medical field both in hard and soft tissue [81].

Triclosan was also used in another drug-delivery system developed by Kaffashi et al. [128]. The antibiotic was loaded in PLA nanoparticles that were in turn added to a PCL matrix to obtain a bionanocomposite that provides a slow and controlled release of the drug. PLA could efficiently encapsulate triclosan particles, burst release was diminished, and the antibacterial property of the sample was extended to up to two years by virtue of the polymer's higher glass transition temperature and more rigidity compared to PCL. A good dispersion of the nanoparticles was also achieved in the PCL matrix by setting the melt mixing temperature below than the melting point of the nanoparticles and fibers

with a diameter between 40–60 μm were obtained, making them a good candidate for soft connective tissue engineering and long-term drug delivery.

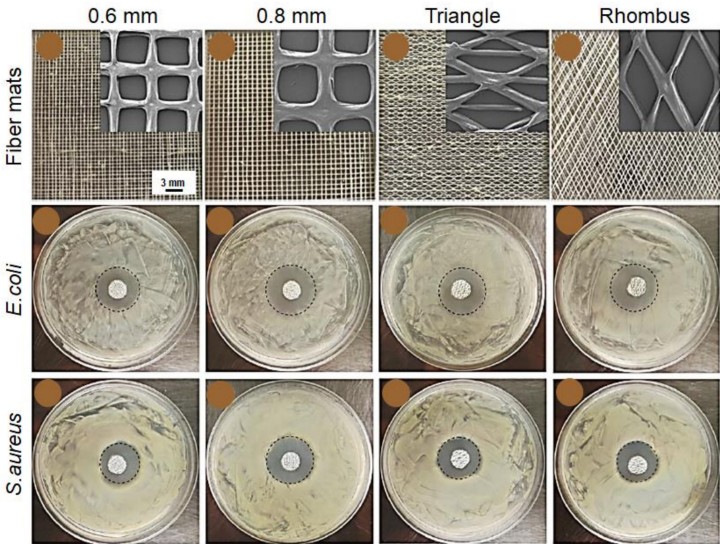

**Figure 8.** PCL/PEG wound dressing fabricated by direct-writing melt electrospinning. SEM images of composite fiber mats with 90:10 PCL/PEG ratio and different geometric structures: Square-shape grid with the spacing of 0.6 and 0.8 mm, triangles and rhombuses. All fiber mat images are of the same magnification. Test of bacteria (*Escherichia coli* and *Staphylococcus aureus*) inhibition zone of fiber mats loaded with ciprofloxacin. Copyright 2018 John Wiley & Sons [116].

Antimicrobial drugs have not only been studied in polyesters. Recently, there has been an increasing interest in polyolefin fibrous membranes containing antibacterial agents for their high efficiency in air and water filtration. Polyolefins are hydrophobic polymers that can be colonized by harmful bacteria [129–131]. Rahman, et al. [21] developed a PP and high-density polyethylene (HDPE) blend loaded with chlorhexidine dihydrochloride (CHDH) fibrous membrane for medical and filtration applications. CHDH is a common ingredient in oral rinse and toothpastes due to its antibacterial, antifungal and antiviral properties [132]. The main advantage is that its toxicity to mammals is very low [133]. In this experiment, the fibrous membrane had a porosity of 77%, with a mean flow pore size of 10 μm. CHDH-loaded PP/HDPE melt electrospun fibers proved to be effective and efficient as the bacterial cell reduction was that of 99.9% in only 4 h [21].

Drug delivery of antithrombotic and antithrombogenic agents like dipyridamole (DPD) is also possible. DPD was blended with PLA and polyhydroxy butyrate (PHB) in a work by Cao, Liu, et al. [134] to obtain a melt electrospun drug delivery system. PHB is a nontoxic polymer obtained from a natural source. It presents biocompatibility and controlled biodegradability, which makes it suitable for biological applications [135]. However, they can be limited by its brittleness and poor thermostability. Nevertheless, as it has been previously mentioned, polymer blends and additives may enhance the properties of materials at different concentrations. In this case, two different proportions of PLA/PHB were tested (9:1 and 7:3) and whilst the first showed a higher resistance to polymer hydrolysis, the latter presented a rate of diffusion transport of DPD twice as high as the 9:1 system. Different content of DPD was also tested and it was observed that the addition of the drug reduced the melting temperature of the blend. DPD-free fibers showed a smooth surface with a relatively constant diameter, whereas fibers with DPD had a rough surface with varying diameter within a single fiber. It should be noted, though, that the drug release profile is not constant because both polymers of this blend are sensitive to hydrolysis and so the drug was released to the medium not only by diffusion but also because of surface degradation of the fibers. Thus, this system should be further studied.

One of the most recent studies on melt electrospun drug delivery systems is by Xu et al. [136]. They prepared a black plaster/PEG nanocomposite to promote the in vitro release of water insoluble drugs found in black plasters. The matrix of the black plaster plays the role of the drug carrier and contains a paste of vegetable oil and Red Dan, the main component of which is $Pb_3O_4$. Several water-insoluble drugs are found in the paste; one example is Sanguis Draconis (SD) or Dragon's Blood. SD is a resin obtained from the rattan palm *Daemonorops draco* that is used in traditional medicine to prevent osteoporosis [137].

Since drug absorption by skin is not effective, PEG was added to improve the efficiency as the common drug carrier that it is for solid dispersions [138,139]. It was proven in the results that when PEG conformed 85% of the fiber composition, the dissolution rate of SD higher than in traditional black plaster. Moreover, the fibers showed uniform diameter with higher porosity and specific surface area, and presented an excellent degradation of the nanocomposite under physiological conditions [136].

**Table 2.** Pharmaceutical additives in melt electrospinning.

| Polymer | Additive | Drug Activity | Effects in the Process and/or Application | Ref. |
|---|---|---|---|---|
| Eudragit® E | Carvedilol | Beta-blocker | Plasticizers, polymer melting reduced and prevented the drug degradation. | [97] |
| Soluplus® | Indomethacin | Anti-inflammatory | Plasticizers, fast drug release. | [98] |
| PCL | Curcumin | Anticancer | Not affect the fiber morphology, low drug release rate. | [90] |
| | Daunorubicin hydrochloride | Anticancer | Fiber diameters barely fluctuated and smooth surfaces. Localized tumour therapy with slow release rate. | [115] |
| | Triclosan/PLA nanoparticles | Antimicrobial | Fibers diameter of 40–60 μm. Soft connective tissue engineering and long-term drug delivery. Slow and controlled release. | [128] |
| PCL/PEG | Ciprofloxacin | Antibiotic | Wound dressing. | [116] |
| HDPE | Chlorhexidine dihydrochloride | Antimicrobial | Fibrous membrane for medical and filtration applications with porosity of 77% and pore size of 10 μm. | [21] |
| PLA/PHB | Dipyridamole (DPD) | Antithrombotic, antithrombogenic | The drug reduced the melting temperature. Fibers had a rough surface with varying diameter. Drug delivery system. | [134] |
| Black plaster/PEG | Sanguis Draconis resin | Prevent osteoporosis | Traditional medicine. | [137] |

## 6. Melt Electrospun Nanocomposites

Although several investigations focused on the preparation of nanocomposite fibers in solution electrospinning [140–142], the published papers discussing the melt electrospinning of polymer nanocomposites are rare due to the increasing complexity of the system. In this section, some research studies that investigated the influence of nanomaterials on both processing and final properties of obtained fibers are summarized (see Table 3).

One of the benefits of electrospun membranes is the possibility of carrying particular agents for the desired applications. One example would be the photothermal conversion that takes advantage of the materials that are able to absorb light energy and convert it into thermal energy. Applications of such materials may include sterilization, desalination, water evaporation and power generation [143,144]. Furthermore, development of materials capable of floating on water are quite interesting since they allow the heating of only the air-

water interface instead of the bulk water, the generation of a sharper localized temperature, and a more efficient evaporation of water [145]. The system based on tungsten oxide/PLA fiber membrane has got an efficient light-to-heat conversion, great NIR absorption, and high floatability on water due to its high hydrophobicity [146]. Membranes consisting of tungsten oxide/PLA wires have been successfully fabricated by melt electrospinning under specific conditions (i.e., nozzle temperature of 260 °C, electric field of 4 kV/cm, drum rotation speed of 100 rpm, and extrusion rate of 0.013 g/min). Membranes were subsequently heat-treated at 80 °C for 10 min to release the residual stress caused by stretching during the fiber spinning process. Final materials became stable without any evidence of shrinkage under exposition to infrared light irradiation. Nonetheless, even though nanoparticles were dispersed homogeneously in the PLA matrix, the morphology and fiber diameter (i.e., 8–13 μm) did not change with nanoparticle addition, even for a 7 wt.% content.

Poly (vinylidene fluoride) (PVDF) is one of the suitable polymers in the application of lithium-ion separators and gives the opportunity to replace polyolefin systems because of its good electrolyte affinity, excellent thermal stability and desired electrochemical performance [147]. The electrospinning technique can provide nanofibrous membranes with a controllable fiber diameter and high porosity for lithium-ion battery separators, improving the lithium-ion conductivity, electrolyte absorption, and retention rate [148]. Coating of separators with ceramic layers like $SiO_2$, $Al_2O_3$, $ZrO_2$, $TiO_2$, and $CeO_2$, were found to be an effective and economical way to improve the thermal stability and wettability of the separator [149]. Sputtering of the mentioned ceramic particles into melt electrospun three-dimensional PVDF networks, with a smooth surface and a diameter of 3.2 μm, boosted the ion conductivity to 2.055 mS/cm at room temperature—much higher than that of the commercial PE separator. In addition, an improvement in discharge capacity to 161.5 mA.h/g and a retention rate of 84.3% after 100 cycles were reported. This ceramic nanoparticle-coated melt-electrospun PVDF is a promising separator for a high-power and more secure lithium-ion battery using a cost effective and eco-friendly process.

Nanomaterials containing $TiO_2$ have been found to be effective in removal of dyes from wastewater, which draws a lot of attention as it minimizes the threat for human health [150]. Typical dye-removing methods include chemical and physical methods, such as chemical coagulation and flocculation, precipitation, and ozonation [151]. Since titanium nanoparticles are potential materials for decolorization of textile dye effluents, the preparation of a $TiO_2$-functionalized polymeric fabric would be an efficient way to improve dyeing-wastewater decolorization [152]. Melt electrospinning was applied in the preparation of PP fibers containing $TiO_2$ nanoparticles for a photocatalytic application. Prior to electrospinning, titanium nanoparticles and sodium stearate were added into PP to boost the electrical conductivity of the polymer melt and improve the electrospinning process. Optimized process conditions (applied voltage, tip-to-collector distance, and flow rate) provided melt electrospun fibers of pure PP and PP/$TiO_2$ nanocomposite with an average diameter of 700 and 800 nm, respectively. With a 90% of decolorization efficiency, this melt electrospun nanocomposite fabric can be considered as a suitable filter with a good photocatalytic activity.

Silica ($SiO_2$) nanoparticles are amongst the most widely used inorganic materials due to their mild reactivity and good chemical properties. Adding $SiO_2$ nanoparticles in a polymer like PET would provide a nanocomposite fiber with good thermal and chemical stability, excellent mechanical properties, and well dye ability [153]. Nano- and microfibers of PET/$SiO_2$-nanocomposite were successfully prepared by laser melt electrospinning with a fiber diameter range between 500 nm and 7 μm [154]. Laser current and applied voltage significantly changed the morphology of the fiber. At high voltage and low current, spray-like particles and ultrafine fibers could be fabricated.

Nanocomposite fibers based on poly-L-lactic acid (PLLA) and nHAp [59] have also been obtained by melt electrospinning. PLLA containing HAp has been widely used in biomedical application thanks to the its biocompatibility and osteoconductivity, and also

because it improves some properties of the polymer [155]. Uniform fibers were obtained, and specifically the addition of a 3 wt.% of nHAp resulted in a decrease of the fiber diameter from 7.1 μm to 4.5 μm, but with a change from smooth to a rough surface. The hydrophilicity of the mats was also enhanced to a certain extent due to the hydrophilic nature and ionic affinity of nHAp particles.

Nanoclays are also widely employed to improve the properties of the matrix polymer. Fibrous nanocomposites based on nylon 6 (PA6) and montmorillonite (MMT) have been prepared by melt electrospinning [156]. The phyllosilicate MMT, when added in the range of 1–10 wt.% into PA6, leads to an increase in the Young's modulus of a single melt electrospun fiber by a factor of 2 relative to the fibers of the same diameter prepared from the pure polyamide. A good dispersion of MMT into the polymer media is mandatory, hence melt processing is used to prepare the nanocomposite with desired dispersion state prior to melt electrospinning. It was reported by Malakhov et al. [156] that an increase in the MMT concentration leads to an increase in the number of charge carriers, which mobility increases with temperature. Both factors lead to an increase in the electrical conductivity of the melt to $10^{-6}$ S cm$^{-1}$ for a MMT content of 3 wt.%, which represents a factor of 4 compared to the initial polymer. Nevertheless, viscosity also increased and therefore thicker fibers were usually obtained (e.g., diameters of 8.1 μm and 12 μm were determined when the MMT increased from 0 to 3 wt.%, respectively). Control of viscosity by increasing the processing temperature was not feasible due to polymer degradation evidences. Fibers showed also an increased roughness surface when the MMT was increased.

Hyperthermia is an effective method for cancer treatment that can be applied by direct injection of magnetic particles into tumour tissues. Nevertheless, it is also necessary to use a biopolymer as a support since the direct injection of particles presents some disadvantages like leakage from the tumour site into the body. Composite nanofiber membranes could be prepared and then be attached to the surface of tumour tissues [157,158]. In this regard, direct in situ electrospinning onto the surface of the tumour tissue may be a suitable method to prevent the shedding of the composite fiber membrane and increase also the fitting of the composite fiber to the tumour [159]. By using a homemade self-powered portable melt electrospinning apparatus (see Figure 9 for the detailed process), a PCL/Fe$_3$O$_4$ membrane constituted by fibers with uniform diameter around 4–17 μm could be fabricated [160]. Fe$_3$O$_4$ magnetic nanoparticles affected the viscosity and conductivity of the melt, which in turn affected the electrospinning process. A small number of nanoparticles provided a stable spinning process with smooth and well-distributed fibers, being possible to achieve and control a 5 μm fiber diameter at a spinning distance of only 5 cm. High nanoparticle content caused an increase in this fiber diameter up to 15 μm and a gradual roughing of the fiber surface. In vitro hyperthermia tests demonstrated that these magnetic fiber membrane composites produced by means of hand-held melt electrospinning devices had a high potential for magnetic hyperthermia treatments. In addition, this portable melt electrospinning device may also be easily extended to wound dressing and haemostasis using nanocomposites with excellent heating efficiency and thermal cycling characteristics.

In bone tissue engineering applications, ideal pore sizes have been reported to range between 100 and 400 μm [161], values that cannot be obtained by a simple solution electro-spinning [162]. A novel PCL/nHAp nanocomposite fibrous scaffold has been developed by Abdal-hay et al. [163] by means of direct melt-electrospinning writing. Figure 10 shows a scheme of this process, which is able rendering continuous orderly stacked fibers according to a 0–90° pattern. and rough fibers after the incorporation of nHAp. This addition increased significantly the roughness of the PCL fibers and also their diameters (e.g., from 10.79 μm to 16.84 and 20.46 μm for 3 and 7 wt.% of nHAp, respectively). Positive cell/material interactions were demonstrated from the study of osteoblasts activity. Furthermore, viability tests indicated a great potential of the new scaffolds for tissue engineering applications.

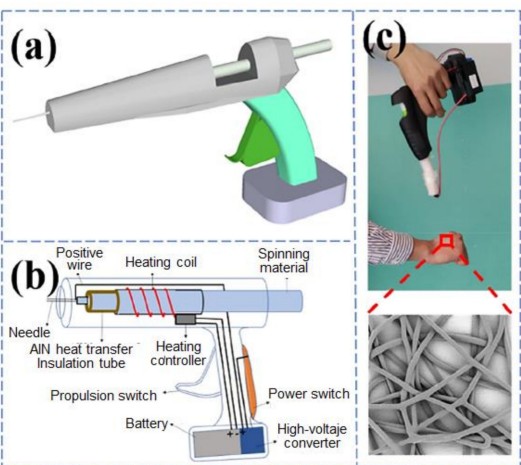

**Figure 9.** (**a**) 3D-model diagram of hand-held melt-electrospinning apparatus. (**b**) Schematic diagram of hand-held melt-electrospinning apparatus. (**c**) Melt electrospinning schematic by hand-held melt-electrospinning apparatus. Copyright 2020 Elsevier [160].

## 7. Melt Electrospun Polymer Blends

As discussed in melt processing section, addition of one or more polymers to another polymer is an efficient and economical way to improve properties of the targeted polymer. Modification of mechanical properties, glass transition and crystallization temperatures, increase of biodegradability and biocompatibility, decrease of the price and increase of the life time are the main reason that brings polymer blending into the attractive category [164–166]. Some polymer blends are miscible and result in homogeneous structures after mixing, nonetheless, majority of blends are immiscible and the multicomponent materials complicate the system and melt processing due to different chemical/physical structure and viscoelastic behaviour [167]. Since the elongational flow of polymer melt plays an important role in the melt electrospinning process, properties and behaviour of polymer blends needs to be rationalized. Even if a lot of studies on polymer blends have been carried out with solution electrospinning, there are only a few that addresses on melt electrospinning of blends. In this scope, information about the properties of polymer melt during the melt electrospinning is scarce, so this could be an open area for research studies.

As discussed before, one polymer used widely in tissue engineering and scaffold construction is PCL. This polymer is attractive due to applications in bone, cartilage and nerve regeneration among other tissues in the body and showed slow degradation rate [168]. Although solution electrospun was successfully applied for preparation of PCL fibers, the addition of an amphiphilic polymer can extend the functionality of the material [167]. In addition, hydrophobic drugs can be added into PCL that are released as the material degrades. In this case, the presence of an amphiphilic material, such as poly (ethylene glycol)-block-poly($\varepsilon$-caprolactone) (PEG-*b*-PCL), allows the delivery of hydrophilic compounds from the fiber and influences the degradation kinetics and hydrophilicity of PCL [169]. Additionally, using a low-molecular-weight PEG-*b*-PCL copolymer is expected to decrease the average fiber diameter by increasing the jet draw ratio, due to the introduction of polar PEG segments, as well as to reduce the viscosity of the polymer melt. From a viscoelastic perspective, mixing a high-molecular weight polymer (i.e., PCL) with a low-molecular weight polymer (i.e., PEG-*b*-PCL) complicates the blend system regarding the difference in physics of molecules as well as on chemistry. Pure PEG-*b*-PCL copolymer melt electrospinning did not result in consistent, continuous and uniform fibers due to its low molecular weight, insufficient macromolecular entanglements and Rayleigh instabilities. By contrast blends of PCL and PEG-*b*-PCL, for some parameter combinations and certain weight ratios of the two components, were able to produce continuous fibers significantly thinner (i.e., with an average diameter close to 2 μm) compared to pure PCL [167].

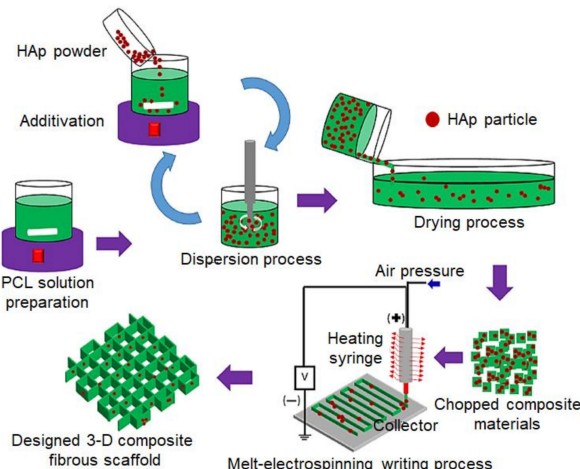

**Figure 10.** Schematic diagram of the 3D fibrous-composite scaffolds fabricated by the melt electrospinning writing process. Copyright 2020 Elsevier [163].

PEG is also considered as an effective plasticizer for PLA-based systems. Thus, PLA/PEG blends with improved cell affinity, biodegradability, as well as physical and mechanical properties, have been found good candidates for both industrial and biomedical applications [7,142]. The effects of spinning parameters were investigated by Nazari and Garmabi [170–172]. Increasing the PEG content, temperature, and voltage decreased the diameter of fibers since it influenced on the viscosity, the quenching, and the stretching of polymer jets. While pure PLA was not spinnable at the temperature of 200 °C, PLA/PEG blends were easily spun, achieving the lowest fiber diameters of 3–6 μm with 30 wt.% of PEG at an applied voltage of 70 kV. It was also found that viscoelastic parameters such as zero-shear viscosity and relaxation time as indicators of elasticity can give relevant information to optimize the processing. Addition of PEG into PLA resulted in enhanced polymer-chain mobility and disentanglement. To predict the miscibility at the melt-electrospinning temperature, interfacial tension between the two phases was calculated from rheological data, which predicted a good miscibility between both polymers. However, it was found that at the higher critical concentration of PEG (i.e., 30%), not enough elasticity existed causing the decrease of the melt electrospinnability.

**Table 3.** Nanoparticle additives in melt electrospinning.

| Polymer | Nanoparticle | Effects in the Process and/or Application | Ref. |
|---------|-------------|-------------------------------------------|------|
| PLLA | nHAp (3 wt.%) | Hydrophilic matrix. Biomedical applications due to its biocompatibility and osteoconductivity. | [59] |
| PLA | $WO_3$ (7 wt.%) | Efficient light-to-heat conversion, great NIR absorption, high hydrophobicity. | [146] |
| PP | $TiO_2$/Sodium stearate nanoparticles | Dyeing-wastewater decolourization, photocatalytic applications. | [152] |
| PET | $SiO_2$ nanoparticles | Good thermal and chemical stability, excellent mechanical properties, and well dye ability. | [153,154] |
| PCL | $Fe_3O_4$ magnetic nanoparticles | Magnetic hyperthermia treatments (cancer treatment), wound dressing and haemostasis. | [160] |
| | nHAp | Increase roughness and diameter of the fibers. Tissue engineering applications. | [163] |
| PA6 | MMT (montmorillonite) (1–10 wt.%) | Viscosity, Young's modulus and electrical conductivity increased. | [156,165] |

Dalton et al. [39] evaluated the effect of viscosity-reducing additives on the phenomenological properties of iPP in the melt electrospinning process. Diameter of fibers could be reduced from 35 μm to 840 nm. Ultrafine fibers of iPP/styrene–acrylonitrile (SAN) copolymer blends have been also obtained by melt electrospinning [173]. Morphology and distribution of SAN in the blend logically varied with composition as well as its nucleating activity and the structure of the electrospun fibers. Even though the addition of SAN resulted in fiber diameters in the rage between 5 and 10 μm and a narrow distribution, more spindle-shaped SAN particles were produced when SAN content was increased. Separated SAN particles joined at the highest concentrations and even formed filaments along the fiber axis (trends shown in Figure 11a–d). In the melt state, SAN particles were efficiently stretched and, as a result, the specific surface area changed. Regarding these observations, new morphologies of polymer blends can be favoured by melt electrospinning because of their faster stretching and cooling processes with respect to other techniques. It was also demonstrated that by using the dry mixing or melt blending prior to electrospinning, it was possible to tune the morphology by modifying the process conditions and/or the material combinations.

Recently, Zakaria and Nakane [61] added polyvinyl butyral (PVB) into iPP and the prepared film was processed using laser melt electrospinning. PVB is a random and amorphous copolymer is copolymer of vinyl butyral, vinyl alcohol and vinyl. The amorphous character and polarity of PVB facilitate the processing of the blend system onto uniform and ultrafine fibers. The laser melt electrospinning system works with a nozzle-free melt-electrospinning with a line-like $CO_2$ laser melting device. As a heterogeneous nucleating agent, PVB improved the crystallinity of PP. The increase of the PVB content also provided finer fibers with a high production rate due to the decrease on viscosity and an enhancement of both of surface adhesion properties and polarity (see Figure 12a). Internal dispersion was also improved as well as thermal properties, which facilitated the drawing of the fiber during the electrospinning process. In other words, since the PVB is polar and immiscible with PP, the polarity of the blends increased and, as a result, a strong electrostatic attractive force between the Taylor cones and grounded collector was created. To fabricate PP nanofibers, a post-processing step consisting on the removal of PVB by dissolving it in ethanol was carried out. PP nanofibers with a diameter as low as 181 nm could be obtained by using a PVB content equal or higher that 60 wt.%.

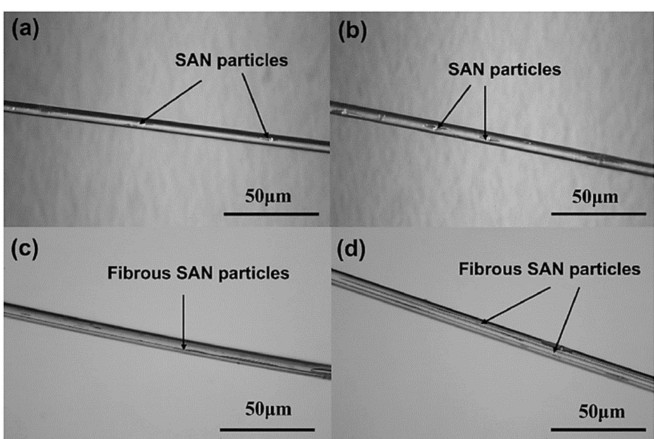

**Figure 11.** Optical micrographs of melt electrospun fibers obtained from SAN and iPP blends. The content of SAN in each image is 0.5 wt.% (**a**), 1.0 wt.%(**b**), 2.0 wt.% (**c**), and 5.0 wt.% (**d**). Copyright 1969 John Wiley & Sons [173].

Poly (ethylene-*co*-vinyl alcohol) (PEVOH) is another polymer that was selected to facilitate the processing of PP fibers. Core (PEVOH)–shell (PP) nanofibers and hollow PP nanofibers have been manufactured from PP/EVOH/PP three-layer films using laser melt electrospinning [174]. The fiber structure depends on the melt flow index (i.e., viscosity).

A suitable core-shell structure can be obtained, as shown in Figure 12b–d) as long as the MFI of PP is higher than that of EVOH. With this information, a series of new high-performance nanofibers from engineering plastics can be developed using melt-electrospinning.

## 8. Relevant Applications of Melt Electrospinning

Similar to the fibers obtained by solution electrospinning, melt electrospun fibers can be appealed in several applications including filtration and separation, food packaging, tissue engineering scaffolds, etc. Especial interests concern industrial scale production and the possibility to avoid solvents [9,15,16]. However, control of fiber size and porosity brings, as previously discussed, some challenges in each melt electrospinning technique and polymeric systems that may be also dependent on the specific application.

### 8.1. Filtration and Separation

Micro/nanofiber membranes are potential systems for water (fresh or waste) and air filtration, petrochemical, electronic, fuel cells, and clean energy systems in industrial scales. However, novel, efficient and long-time usage systems are still being developed in filtration and separation applications. Nanofibrous materials have some advantages that are not present in commonly used materials, such as high surface energy, porous structure with controllable small pore size, and relatively high strength [175]. Nanofiltration is used as a pre-treatment of waste water before the typical reverse osmosis process. Filtration can operate with a high flux and significantly reduces power consumption and system maintenance costs [176]. Regarding to this large-scale consumption (for both industrial and domestical uses), filters obtained by solution electrospinning should cause a large increase in the production of toxic solvents and the corresponding wastes, which can be avoided by means of fusion electrospinning techniques. Among fibrous systems, polypropylene, poly (ethylene terephthalate), polyethylene and nylon are of high interest in industrial filtration [39,49,177]. As it was explained in previous sections, viscosity modifiers and improvement of polarization are fundamental to properly process these polymers, especially PP, into fine and continuous fibers. It was found that gas-assisted melt electrospinning allowed to get PP fibrous membranes with a diameter of 600 nm and a porosity of 99.4%, which rendered much higher performance compared to traditional webs produced for water treatment applications [178].

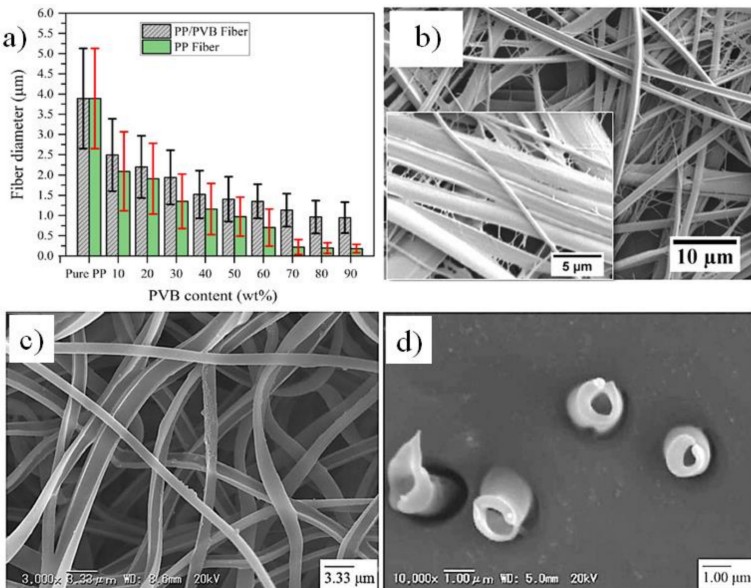

**Figure 12.** Comparison of PP/PVB and PP (after PVB removal) fiber diameters (**a**) and SEM images of PP fibers after PVB removal (**b**). (**c**,**d**) core–shell structure of PP/EVOH based fibers after EVOH removal. Copyright 2019 John Wiley & Sons [61]. Copyright 2018 John Wiley & Sons [174].

Membranes constituted by polyamide (PA) fibers have great interest for filtration of suspended particles from gases, in particular, air. The filters with thicker fiber diameters (i.e., pure PA) showed less efficiency, 21%, while modification of PA with the SS additive increased efficiency to 98% due to its better packing density and surface chemistry [76]. Different commercial polyamide 12 (PA12) and block polyether amide filaments were incorporated in a melt-electrospinning process and the highest filtration performance was showed by PA12 with MFI of 132 g/10 min supplied by Evonik Industries AG [179]. The maximum filtering efficiency was found to be 80% fibers with an average diameter of 15 μm. In addition to PA and PP, melt electrospun polyphenylene sulphide (PPS) superfine fibers proved to be a promising and highly efficient material for high temperature dedusting [180]. Prior to melt electrospinning, PP was added to PPS as an additive to provide suitable finer fibers for filtration. By decreasing the fiber diameter to 4.12 μm, the filter efficiency increased to 98.05% which was much higher than the commercial filter bag for the high temperature filtration of flue gas emission.

To sum up, modification of materials and melt electrospinning processes seem to be the efficient ways to provide membranes with filtration coefficients comparable with those of materials prepared via solution electrospinning. In addition to the research studies, several companies in USA and Germany provide nanofiber filters that mostly focus on solution electrospinning. In this area, there is the possibility to replace this process by melt electrospinning by modification of materials using additives and changing the processes as previously discussed.

### 8.2. Food Packaging

Developing new food packaging systems is always interesting to support physicochemical, optical, mechanical and catalytic properties in food industries. Active food packaging provides the ability to control moisture, ethylene and $CO_2$ absorbers, oxygen scavengers and antimicrobial and antifungal agents, while the smart packaging act as a sensor to show temperature, freshness and gas indication [181,182]. There is a growing demand in the production of food packaging materials with a high loading of active agents and highly responsive nature so that materials can release active molecules as a response to environmental conditions. This can be achieved by melt electrospinning. In addition, the small fiber spacings act as a barrier against the bacterial entrance [183,184]. Different active agents including antioxidant, antimicrobial and antifungal agents are shown in Figure 13 together with different electrospinning techniques. Hybrid electrospinning or co and coaxial electrospinning are effective processes to incorporate active agents in polymeric media according an in-situ process [183]. Nonetheless, one of the drawbacks of solution electrospinning for food packaging applications is again the use of toxic solvents, which limits the compatibility of the obtained fibers with food.

Potential of melt electrospinning to prepare active/smart packaging is currently relevant [184]. Moreover, several agents have been effectively incorporated by means of melt electrospinning. Relevant examples correspond to phenolic compounds with antioxidant properties such as carnosol, carnosoic acid, rosmanol, rosmadial, epirosmanol, rosmadiphenol and rosmarinic acid. Essential oils, such as α-pinene, bornyl acetate, camphor and 1,8-cineole have been applied due to their strong antimicrobial properties that are fundamental in packaging industry. Bhullar [184] developed for example a bioactive antimicrobial polymeric microfibrous structure using PCL and a natural extract from the rosemary plant, which is a popular antimicrobial agent. It was found that physical properties such as morphology and thermal stability of pure PCL remained unchanged and that logically the antimicrobial efficacy was increased. Although, challenges of melt electrospinning (e.g., degradation of polymer or added agents during melt processing) limit this eco-friendly technique, researchers provided possible ways to customize the polymer fibers for food packaging applications as discussed in this review.

### 8.3. Biomedical Applications and Tissue Engineering

If melt electrospinning is at the dawning, the technique applied in tissue engineering has to come a long way to see the first sunrays. For this reason, most papers in this field discuss experiments performed in vitro. Very few have reached the in vivo or clinical stages. Nevertheless, tissue engineering and the requirements that engineered constructs must meet have been vastly studied for more than 30 years [185–187].

Generally, tissue engineering is used to repair and/or regenerate damaged tissues and organs [15]. 3D nanofibrous structures, namely scaffolds [188], that mimic the structure and function of the natural extracellular matrix (ECM) [189] are manufactured and, by means of melt electrospinning or melt electrowriting, a wide range of fiber diameters can be achieved as well as well-defined and controllable geometries and porosities.

As the temporary substitutes they are, scaffolds must provide the appropriate microenvironment to promote cell adhesion, migration, proliferation, differentiation and secretion of cells own ECM [190–192], and eventual formation of the tissue or organ [193]. These processes can be induced mechanically or chemically, or by the scaffold structure, which can regulate the cell alignment [194,195]. In mechanical induction, press [196], tensile [197], or shear [198] stress are applied on cells to induce polarization, whereas in chemical induction, growth factors are used to attract cell penetration [199].

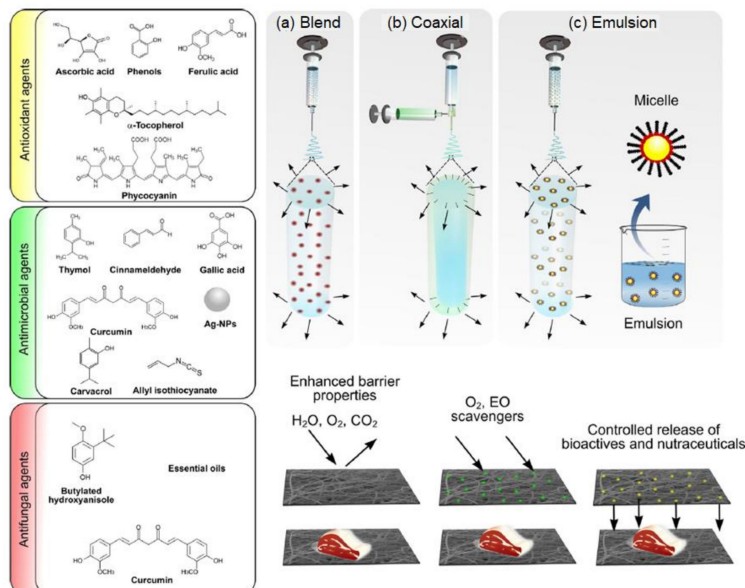

**Figure 13.** The fabrication and performance of functional electrospun materials for food packaging applications. EO, essential oil. Copyright 2020 Elsevier [183].

The design of the scaffold and the selection of growth factors are based on the damaged tissue (e.g., vascular, endothelial, adipose, skin, muscle, cartilage, and bone) that is wished to repair [200]. Moreover, not only are there differences among each type of cell or tissue and how it interacts with the scaffold [161], but cellular invasion also depends on whether the experiment is being carried out in vitro or in vivo, especially due to the vascularization [201,202].

The diffusion limit of oxygen and nutrients within tissues is found to be between 150–200 µm. Most tissues rely on blood vessels for the transport of oxygen and nutrient supply [203,204], hence, control over vascularization is so important in larger tissues, as the nearest capillary should never exceed that distance [205,206]. Vascularization of scaffolds prevents necrosis of tissues and is achieved by spontaneous vascular ingrowth as a response to hypoxia that implanted cells secrete [207]. In vitro cultures can also be supplied with nutrients and oxygen in certain bioreactors [208]. However, not everything is "a piece of cake". The capillary growth is limited to tenths of micrometres per day [209] and doing

the calculations, this means weeks are needed to complete the vascularization in implants in the order of millimeters. This could not only lead to hypoxia and an eventual necrosis of the tissue, but also to a non-uniform cell differentiation and thus, a decreased tissue function [210]. Therefore, vascularization is a current challenge that needs to be overcome or circumvented [211].

Melt electrospinning and preferably melt electrowriting, most inherent problems could be tackled as scaffolds design parameters, including structure, shape, size, porosity [212], and fiber diameters and spacings [213,214] can be adjusted according to the needs of the tissue in question. This is one of the main advantages of using molten polymers: these are often viscous and nonconductive, avoiding this way electrical instabilities in electrospinning jets. The technique allows a predictable fiber deposition, a layer-by-layer fabrication of scaffolds with defined designs, shapes and thicknesses [202].

In an in vitro study performed by Brown et al. [215], melt electrospun PCL tubular scaffolds were proved to be noncytotoxic and to support cell growth of three different types of cells: primary human osteoblasts (hOB), primary mouse osteoblasts (mOB), and human mesothelial cells. The low melting temperature, long degradation kinetics of PCL and its capability to be blended with other polymers has made it a good material for tissue engineering applications. Furthermore, to mimic the bone microenvironment, the scaffolds can easily be coated with a layer of calcium phosphate (CaP). Scaffolds seeded with human cells presented a >90% cell viability and those seeded with mOBs showed the formation of a mineralized matrix after 4 weeks of culture on the scaffold fibers and inside and across the pores [215]. Further investigations by Zaiss et al. [212] led to the manufacturing of PCL scaffolds with an average fiber diameter of 15 μm and an average pore size of 250–300 μm, which were again coated with a CaP layer. In vitro ovine osteoblasts proliferation on such scaffolds was studied. Results showed a cell viability above 90% after 20 and 40 days of culture, a completely covered scaffold surface after 40 days, and the secretion of ECM.

Recent studies seem to demonstrate that thicker fiber diameters (i.e., in the range of a few micrometres) have an increased proliferative potential compared to nanofibers. However, smaller-sized fibers promote a better cell adhesion due to the high surface area that nanomaterials possess [216–218].

In between, Lee et al. [219] prepared fiber mats in another application for bone tissue regeneration using PLA. This polymer has been extensively been applied in many biomedical applications because it is biocompatible, biodegradable and renewable [220]. Furthermore, it presents a mild foreign reaction [221] without inflammatory reaction [222]. Compared to the performance of the same experiment with a solution-electrospun fiber mat, melt electrospun fibers showed better results in cell viability and differentiation.

Kim et al. [200] took a step forward and carried out an in vivo study, proving that it is possible to regenerate rabbit calvarial defects using hybrid nanofiber scaffolds of silk fibroin (SF) and poly(glycolic acid) (PGA). Silk fibroin is a natural biopolymer with excellent biomedical and mechanical properties, including tensile strength, biocompatibility, hydrophilicity and biodegradability, which makes it a good candidate for bone tissue engineering applications [223]. It can also be combined with other materials to modify certain properties. In this case, PGA was chosen mainly for its biodegradability and thermoplastic properties, which enhance cell proliferation. The scaffolds were inserted in rabbit damaged crania and the analysis performed after 4 and 8 weeks revealed that SF-PGA hybrid scaffolds had indeed promoted bone tissue regeneration [200].

Another application including bone tissue engineering is the periodontal tissue regeneration. Two relevant studies included in vivo experiments. First, Costa et al. [224] synthesized a PCL composite with β-tricalcium phosphate that was later coated with CaP. The scaffold was then seeded with osteoblasts and cultured in vitro for 6 weeks before being complemented with periodontal ligament cell sheets, attached to a dentine block and implanted into athymic rats. Results confirmed that the tissue was integrated between the bone and the periodontal ligament compartments as there were high levels of vascularization and a tissue with a structure similar to that of native periodontal tissues was

formed. The second study was performed by Vaquette et al. [225] using highly porous PCL scaffolds that were seeded with cells from three different sources: gingival cells (GCs), bone marrow-derived mesenchymal stromal cells (Bm-MSCs), and periodontal ligament cells (PDLCs). After the healing process of a periodontal ovine model for 5–10 weeks, the implants with Bm-MSCs and PDLCs showed much better results, indicating that periodontal regeneration was feasible.

Bones are composed of cortical bone and trabecular bone at different ratios depending on the specific bone. The fibrous tissue that surrounds the outer cortical bone surface is called periosteum and it contains blood vessels, nerve fibers, osteoblasts and osteoclasts, hence it plays a very important role in fracture repair [226] and thus, in bone regeneration.

Baldwin et al. [227] combined a star-shaped PEG heparin hydrogel system loaded with human umbilical vein endothelial cells (HUVECs) with a medical grade PCL tubular scaffold seeded with human Bm-MSCs at an attempt to regenerate the periosteum in mice. These two types of cells were used to mimic both the vascular and osteogenic functions. HUVECs increased the vascularization. However, 30 days after the implantation in vivo, Bm-MSCs retained the undifferentiated state, which could be due to the properties of the hydrogel. The negative charge of heparin caused a slow release of growth factors. Furthermore, host cells gradually replaced human cells. This could be attributed to the species-specific differences between the host and donor cells.

PCL has been also applied to manufacture scaffolds for oral hard and soft tissue, i.e., bone and mucosal regeneration. Fuchs et al. [228] developed multi-layered membranes for an in vitro study. These membranes, composed of specific surface scaffolds for each tissue as well as bacteria-tight core membranes for an infection-free guided tissue regeneration, were manufactured via melt electrowriting. Since scaffolds with square pores have been previously reported to promote osteoblast and endosteal cell growth [229], a box-structured surface scaffold was designed for the bone side. As for the mucosa tissue, triangular pores were chosen. The result after fusing both scaffolds with the bacteria-tight core membranes was the obtainment of individual bi-layered membranes. Cell growth was finally evaluated by seeding osteoblasts on the box-shaped membrane, and keratinocytes and fibroblasts on the membrane with triangular pores. Results showed these membranes could represent a potential method for bacteria-free wound healing and guided tissue regeneration in oral and maxillofacial surgery.

Other applications in tissue engineering where melt electrospinning is employed include articular cartilage regeneration. Cartilage is composed of a dense ECM with highly specialized chondrocytes—the main composition of which is water, collagen, proteoglycans, other non-collagenous proteins and glycoproteins in lower amounts (Figure 14a). The tissue lacks of blood vessels, lymphatics and nerves. Its primary function is the transmission of loads with a low coefficient of friction, hence its surface must be smooth and lubricated [230].

Bas et al. [231] developed a negatively charged star-shaped PEG/heparin hydrogel combined with a medical grade PCL melt electrospun fibrous network. The resulted fibrous-reinforced hydrogel exhibited good mechanical properties and had a microenvironment analogous to the negatively charged proteoglycan matrix of native cartilage, which promoted in vitro neocartilage formation when seeded with human articular chondrocytes (Figure 14b).

Tendons and ligaments are tissues that connect bone to muscle and bone to bone, respectively. Injuries of these tissues are both common and serious, and their repair represent a current challenge due to their poor natural healing response [232]. For this reason, efforts are being made to find a treatment that helps their regenerate. That is where techniques like melt electrowriting and melt electrospinning may play a significant role in tissue engineering. Both tissues possess similar ultrastructure and physiology and perform similar functions [233]. The main composition of tendons and ligaments are fibrillar collagens of different types, type I is the predominant one in tendons and contains regions of $\alpha$-helices [232] that led to an elastomeric character.

In an in vitro approach, Hochleitner et al. [234] produced scaffolds via melt electrowriting to emulate the biomechanical behavior of tendons and ligaments. The material used was a copolymer of PCL and acryloyl carbonate p(CL-*co*-AC) to manufacture aligned, crimpled, and mechanically robust elastomeric scaffolds. These were proven to be cytocompatible after performing a test with murine cells. Cell adhesion was also demonstrated with human mesenchymal stem cells (hMSC). Thus, these constructs entail a promising approximation in tendon and ligament tissue engineering.

One of the most recent works concerning melt electrowritten fibers applied in tissue engineering was carried out by Gwiazda et al. [235]. The aim was to study the cell organization based on the fiber arrangement, which is crucial in orthopedic applications in ligament tissue engineering as a high degree of cell alignment is found in the native tissues. Therefore, three different patterns of porous PCL scaffolds were tested (aligned, crimped, and random). The aligned-fibrous scaffold was the only one that permitted cells to orientate towards the fiber main direction. Furthermore, for the development of a functional scaffold with an application in the anterior cruciate ligament and an eventual bone-ligament-bone construct, this scaffold was combined with a melt electrospun compartment that was seeded with osteogenically induced hMSC. Experimental results showed that fiber organization is capable of inducing cell alignment and that cell-seeded bone-ligament-bone structures have increased mechanical resilience and elasticity even though the mechanical properties were lower than those of native tissue.

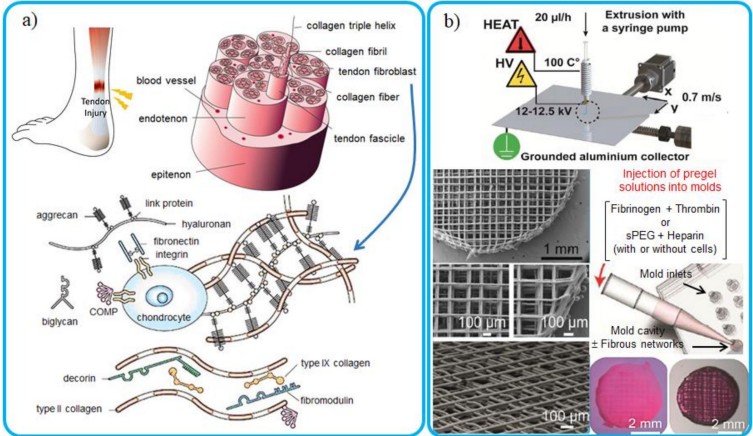

**Figure 14.** Scaffold implantation and cell therapy as approach for repair of tendon injuries. (**a**) The hierarchical architecture of tendon: Collagen triple-helices self-assemble into fibrils. Bundles of fibrils form fibers, which constitute tendon fascicle. Tendon fibroblasts (tenocytes) reside between collagen fibers. Fascicles are wrapped by endotenon, a layer of connective tissue containing blood vessels, nerves and lymphatics. Multiple fascicles are further surrounded by another connective tissue layer, epitenon, to form the tendon tissue [232]. Extracellular matrix (ECM) of cartilage: Three classes of proteins exist in articular cartilage: collagens (mostly type II collagen); proteoglycans (primarily aggrecan); and other noncollagenous proteins (including link protein, fibronectin, cartilage oligomeric matrix protein) and the smaller proteoglycans (biglycan, decorin and fibromodulin). The interaction between highly negatively charged cartilage proteoglycans and type II collagen fibrils is responsible for the compressive and tensile strength of the tissue, which resists load in vivo. Abbreviation: COMP, cartilage oligomeric matrix protein [230]. (**b**) Biofabricated soft network composites for cartilage tissue engineering: melt electrospinning writing (MEW) allows the design and fabrication of medical grade PCL fibrous networks. A soft matrix of hydrogel to emulate the proteoglycan matrix of articular cartilage was loaded into fibrous networks [231]. Based on [230–232].

Cardiac tissue engineering is another relevant application of melt electrospinning. Like other tissues, the heart presents a low healing response after a cardiac injury and is usually associated with a high mortality rate [236]. Attempts have been made to regenerate the tissue with biomaterials, but the current techniques present a main drawback:

the impossibility to manufacture fiber-organized structures and mimic the mechanical environment of native cardiac tissue. This issue can be overcome by melt electrospinning as it allows a scaffold fabrication with aligned fibers. Castilho et al. [54] prepared a blend of poly (hydroxymethylglycolide-co-ε-caprolactone) (pHMGCL), a copolymer that presents higher hydrophilicity compared to PCL, and PCL (pHMGCL/PCL) via melt electrospinning. Additionally, its degradability rate could be tuned, and the fibers could be further functionalized. The melt electrowritten rectangular pattern approximated to the broad mechanical properties of native cardiac tissue. This scaffold promoted the seeded cardiac progenitor cell (CPC) alignment along the fiber main direction of the construct. Moreover, after 7 days of culture, the survival rate was estimated to be 99.9%. However, the pores in the structure were not uniform and so this issue should be addressed in further works as well as improving conductivity of pHMGCL, which has been proved to positively influence CPC behaviour.

In another work, Castilho et al. [237] designed a PCL scaffold with hexagonal microstructures to repair cardiac tissue as a treatment for ischemic heart disease and heart failure. To evaluate its performance, human induced pluripotent stem cells-derived cardiomyocytes (iPSC-CM) were encapsulated in a collagen-based hydrogel and were then seeded on this scaffold. Compared to fiber scaffolds that had been produced up until then, this one appeared to deliver 40 times more elastic energy, showed a 1.5-fold beating rate, an enhanced iPSC-CM maturation and cell alignment, and sarcomere content and organisation. Moreover, an in vivo test using a porcine model was performed to demonstrate the potential of minimally invasive application by creating a functional myocardial patch and injecting it via catheter-like tubing into a beating porcine heart. Even though spontaneous contractions were observed 30 min after the injection, the beating decreased after 2 days. Similar results were observed in vitro after 2 and 5 days (Figure 15).

In another cardiac tissue engineering application, Liao et al. [238] fabricated a bilayered scaffold to promote tissue growth at the interface between a suture-less inflow cannula to be implanted without bypass and in the left ventricle of the myocardium. The first layer of the scaffold was a silicone based layer to mimic the seal, and the other layer was a melt electrospun PCL scaffold with the function of integrating tissue as human foreskin fibroblasts were seeded onto it. At day 1 of culture, the scaffold showed a >95% cell viability and the percentage of dead cells did not significantly increase over the time course, which indicated no cytotoxic effects.

Yet in another heart tissue engineering study, a scaffold mimicking a heart valve was developed by Saidy et al. [239]. The importance of heart valve tissue engineering lies on the more than 5 million people affected in the US and the 3% sudden deaths in the European Union, annually [240,241]. With respect to the clinically available heart valve prostheses, this scaffold overcomes a few problems, such as supporting tissue remodeling and growth. One of the main components in the tissue is collagen; thus, the microarchitecture of the fiber scaffolds was controlled by melt electrowriting to mimic the crimped structure of natural collagen. Results after cell seeding and culture of human umbilical cord vein smooth muscle cells in the scaffold showed its capability to promote the cells growth, proliferation, and creation of their own ECM [239].

As mentioned early in this section, vascularization is a current challenge in tissue engineering that needs to be overcome or circumvented. A possible solution could be the development of scaffolds that stimulate vascularization by their structure. Bertlein et al. [211] synthesized a highly structured PCL scaffold by melt electrowriting. Human umbilical vein endothelial cells (HUVEC) were seeded onto the scaffold and cultured for one day before adding normal human dermal fibroblasts (NHDF) that had been previously coated with a fibronectin and gelatine nanofilm, both ECM-derived components [26]. After a 3-day incubation period, capillary-like structures were found to be oriented along the fibers. Additionally, secretion of the angiogenic factor (VEGF) from NHDF and the consumption of VEGF by HUVEC were demonstrated. Moreover, VEGF secretion from the tissues was

higher on the scaffolds than on the controls, which implies the generation of neovascular tissue [211].

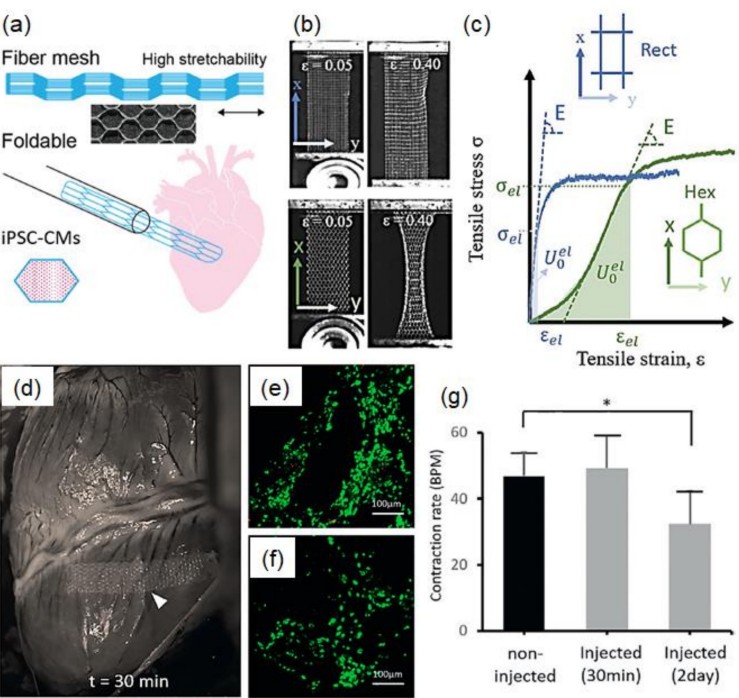

**Figure 15.** Scaffolds for advance functional human myocardial tissue formation. Representation of the workflow and fabricated microfiber scaffolds. (**a**) Designed hexagonal microstructure as a 3D fiber scaffold. The combination with iPSC-CMs combination has in vivo application through minimally invasive delivery. Additionally, optical image of the fabricated scaffolds: detail of microstructure with hexagonal cells (with a side length of 400 μm) composed of multiple stacked aligned microfibers. Images acquired from top perspective. (**b**,**c**) Mechanical behavior of the fiber scaffolds under tensile loading. Corresponding deformed microstructures at different strain levels for both microstructures: top, rectangular, and down, hexagonal microstructures (**b**). Representation of the mechanical parameters determined from the engineering stress–strain curves, where: $\varepsilon_{el}$ is the elastic limit strain, $U_0^{el}$ is the elastic strain energy density, and $E_T$ is the tangent modulus (**c**). (**d**–**g**) Injectability and in vivo placement of cardiac patch with hexagonal geometry. In-vitro culture of cardiac patch consisting of iPSC-CMs in cardiac-like ECM on large hexagonal scaffolds. Application and shape recovery of cardiac patch on beating porcine heart (**d**). Cell viability of in vitro noninjected (**e**) and injected (**f**) cardiac patches. Spontaneous beating rate of in-vitro noninjected and injected cardiac patches 30 min and 2 d after injection. * Significantly different ($p < 0.05$) (**g**). Copyright 2018 John Wiley & Sons [237].

Another promising application of melt electrospinning is the fabrication of skin replacements as burns and wounds have low regeneration capability. Hewitt et al. [242] prepared a PCL blend scaffold with bioactive milk proteins for potential applications in deep skin tissue regeneration. Whey protein (WP) and lactoferrin (LF) were chosen as the milk proteins for this study, not only because of their bioactivity, but because of their specific properties. For example, WP acts as an antioxidant, anti-tumour, antiviral, and antibacterial agent. Another property WP possesses is the capacity to promote cell proliferation and osteogenic differentiation [243]. WP/PCL blend shows lower mechanical strength than PCL, which suggests that this blend may be suitable for soft tissue engineering applications [244]. On the other hand, LF is thought to have an innate immunomodulatory effect [245]. To investigate if tissue regeneration was feasible using these scaffolds, human adult low calcium temperature (HaCaT) keratinocytes and neonatal human dermal fibroblasts were seeded onto them and incubated for 21 days. Results showed that milk proteins

enhanced tissue regeneration and thus, it makes a suitable tissue engineering construct for full thickness dermal skin wounds [242].

## 9. Conclusions

In this review, research studies focusing on modification of process and materials in melt electrospinning were covered and summarized. Compared to solution electrospinning, melt electrospinning provides as more significant benefits such as the possibility of electrospinning of polymers without suitable solvents, lack of toxic solvents and eco-friendly process for producing cost effective fibers. Besides these advantages, the degradation and difficulty to provide ultrafine fiber for different applications are still challenging. Different new techniques were explained to understand better the performance of materials with other modifying components. Salt, viscosity modifiers, stabilizers and dyes were found as high potential additives in melt electrospinning of both biodegradable and non-biodegradable polymers in terms of improving the melt polarity and viscosity, which influence the elongational flow in the process and facilitate the melt spinning to obtain ultrafine fibers. Since one of the important applications of these fibrous scaffolds is in pharmaceutical area, different additives in biomedical applications also were explained comprehensively. Effects of nanoparticles and other polymer(s) were also investigated although using the nanocomposites in the melt electrospinning process may change the melt viscoelastic properties of polymers and complicate the melt spinning process. Addition of other polymer was also found as useful way to improve both the process and final properties of fibers especially in the case of biodegradable polymers. Modification of mechanical properties, glass transition and crystallization temperatures, biodegradability and biocompatibility together with the possibility to decrease costs and increase the lifetime are the main reasons of polymer blending. In addition, some polymers have different solvents which cannot be processed by solution electrospinning. Extraction of one polymer in the blend fibers is an interesting approximation to provide different morphologies like hallow fibers. As an applicable technique, different potential uses of this eco-friendly process are being successfully developed, being results highly promising in different areas such as filtration and separation, food packaging, biomedical applications in general and tissue engineering in particular.

**Author Contributions:** A.B.-H. wrote Sections 5 and 8 and polished English writing. O.Y. wrote Sections 1–4, 6 and 7. L.J.d.V. and J.P. performed supervision, review and editing of the manuscript. All authors have read and agreed to the published version of the manuscript.

**Funding:** This research was funded by the Spanish Ministry of Economy and Competitiveness for the Project RTI2018-101827-B-I00 and the Generalitat de Catalunya for the grant 2017SGR373.

**Conflicts of Interest:** The authors declare no conflict of interest.

## Abbreviations

| | |
|---|---|
| 1D | 1-dimensional |
| 2D | 2-dimensional |
| ATBC | Acetyl Tributyl Citrate |
| Bm-MSC | Bone Marrow-derived Mesenchymal Stromal Cells |
| CAD | Computer-aided Design |
| CaP | Calcium Phosphate |
| CHDH | Chlorhexidine Dihydrochloride |
| CPC | Cardiac Progenitor Cell |
| DOTP | Di-2-ethylhexil terephthalate |
| DPD | Dipyridamole |
| ECM | Extracellular Matrix |
| eMSC | Endometrial Mesenchymal Stem Cells |

| | |
|---|---|
| FDA | Food and Drug Administration |
| GCs | Gingival Cells |
| HDPE | High Density Polyethylene |
| hMSC | Human Mesenchymal Stem Cells |
| hOB | Human Osteoblasts |
| Hap | Hidroxyapatite |
| HUVEC | Human Umbilical Vein Endosteal Cells |
| HUVETs | Human Umbilical Vein Endothelial Cells |
| iPP | Isotactic Polypropylene |
| iPSC-CM | Human Induced Pluripotent Stem Cells-derived Cardiomyocytes |
| LF | Lactoferrin |
| MEW | Melt Electrowriting/Melt Electrospinning Writing |
| MFI | Melt Flow Index |
| MFR | Melt Flow Rate |
| MMT | Montmorillonite |
| mOB | Primary Mouse Osteoblasts |
| nHAp | Nanohydroxyapatite |
| NHDF | Normal Human Dermal Fibroblasts |
| p(CL-co-AC) | Copolymer of PCL and acryloyl carbonate |
| PA | Polyamide |
| PA12 | Polyamide 12 |
| PA6 | Polyamide 6 |
| PCL | Poly ($\varepsilon$-caprolactone) |
| PDLCs | Periodontal Ligament Cells |
| PDMS | Poly (dimethyl siloxane) |
| PE | Polyethylene |
| PEG | Poly (ethylene glycol) |
| PEG-b-PCL | Poly (ethylene glycol)-block-poly($\varepsilon$-caprolactone) |
| PES | Polyester |
| PGA | Poly (glycolic acid) |
| PHB | Polyhydroxy Butyrate |
| pHMGCL | Poly (hydroxymethylglycolide-co-$\varepsilon$-caprolactone) |
| PLA | Polylactic acid |
| PLLA | Poly-L-lactic acid |
| PMMA | Poly (methyl methacrylate) |
| POP | Pelvic Organ Prolapse |
| PP | Polypropylene |
| PPS | Polyphenylene sulphide |
| PEVOH | Poly (ethylene-co-vinyl alcohol) |
| PVB | Polyvinyl butyral |
| PVDF | Poly (vinylidene fluoride) |
| SAN | Styrene-acrylonitrile |
| SD | Sanguis Draconis |
| SF | Silk Fibroin |
| SM | Sodium Myristate |
| SO | Sodium Oleate |
| SS | Sodium Stearate |
| TPU | Polyurethane |
| VEGF | Angiogenic Factor |
| WP | Whey Protein |

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
