# Peer review of "Melt Electrospinning of Polymers: Blends, Nanocomposites, Additives and Applications"

_applsci, doi:10.3390/app11041808_

Round 1

Reviewer 1 Report

The review manuscript described the polymer blend materials and nanocomposites used in the melt electrospinning applications. In the light of a number of previous review articles focused in the similar subject area, this manuscript includes more recently published information as the update and the summary of new techniques and devices developed to provide as better tools for understanding and achieve prospective morphology and performance of polymeric electrospun-nanofiber materials in the presence of additives. It is comprehensive and a well-organized account. Accordingly, the manuscript is publishable after minor typo revisions. These suggestions are given below.

1). There are quite a number of words in almost every page with unnecessary “hyphen” between characters. They need to be corrected.

2). Many format typing errors needed to be checked throughout, for examples, at line 489 of page 14, lines 664 and 693 of page 19, lines 706, 716, 721, 723, 727, 731, 732, 735,  of page 20, line 773 of page 21, ….. etc.

Reviewer 2 Report

Overview and general recommendation:

This paper reviews the principles of melt electrospinning with a focus on additives and blends of polymer melt, its effect on melt electrospinning and different applications of this technique. Authors did a good job connecting the different topics and the flow is good. However, some re-organization of the topics as well as referral to most resent melt electrospinning techniques would enhance the overall deliverable.

Comments for the authors:

  • Section 2 can be summarized and merged with section 5. Otherwise, some connection to the topic of the paper, melt electrospinning, is needed here.
  • Section 3: Where you mentioned that hot drawing improves the chain orientation, how about cold drawing? Which is very common in melt spinning.
  • While this is a holistic paper reviewing melt electrospinning processes and there is a specific section related to melt electrospinning techniques (section 4.2), referencing more recent free surface melt electrospinning techniques as an emerging technique (to scale up the process) would complement the work. To get a recent overview see these 2020 papers:

https://doi.org/10.1021/acsapm.0c01082

https://doi.org/10.1016/j.polymertesting.2020.106865

  • In section 4.1 the diameter of the spinneret is listed as a factor inhibiting thin fiber formation, however viscosity is the contributing factor and spinneret diameter only controls the clogging issues. This should be reconsidered.
  • In section 4, melt electrospinning also needs a proper ventilation system to exhaust the gas released by the polymer melt. This sentence should be revised.
  • Section 4.1: Melt electrospinning conditions could be listed as sub-topics an aside from section 4.

Reviewer 3 Report

This manuscript reviewed the processing techniques, processing parameters, additives, polymer blends and applications of melt electrospinning. It provides a nice overview to a novice reader in the field. Some comments are suggested.

  1. There are many improper hyphenations. Please check the full text and remove it. e.g. Line 16 reduction, Line 22 biomedical and line 95 production and so on.

  1. Authors need a table summarizing some typical productions of melt electrospinning. Some factors that they can use for summary (not an exhaustive list): materials, additives, processing conditions, fiber diameters, applications etc.
